# A long non-coding RNA *Leat1* mediates the hormone responsiveness of *EfnB2* during male urogenital development
Deidre Mattiske [1,5], Pascal Bernard[1,5], Paul E. Gradie[1,5], Richard R. Behringer[2,3], Paul A. Overbeek[3], Rachel J O'Neill [4], Tiffany Phillips[1], Melanie Stewart[1], Neil Youngson[1], Gerard Tarulli [1] & Andrew J. Pask [1] ✉

The novel long non-coding RNA (lncRNA) *Leat1* is extraordinarily conserved in both its location (syntenic with *EfnB2*, an essential gene in anogenital patterning) and sequence. Here we show that *Leat1* is upregulated following the production of testosterone from the developing testis in mice and interacts with *EfnB2*, positively regulating its expression. *Leat1* expression is suppressed by estrogen, which in turn suppresses the expression of *EfnB2*. Moreover, the loss of *Leat1* leads to reduced *EfnB2*, resulting in a severe hypospadias phenotype. The human *LEAT1* gene is also co-expressed with *EFNB2* in the developing human penis, suggesting a conserved function for this gene in urethral closure. Together our data identify *Leat1* as a novel molecular regulator of urethral closure and implicate it as a target of endocrine disruption in the etiology of hypospadias.

Hypospadias is an abnormality of penis development that results in the misplacement of the urethral opening on the underside of the penis. It is one of the most common birth defects in the USA, affecting approximately 1 in every 125 live male births[1]. Hypospadias varies in severity, depending on the location of the urethral opening[2]. Distal hypospadias, where the urethra opens just below its normal location, is the most mild and common form of the disease, accounting for up to 65% of all cases, while the more severe forms, where the urethra opens along the shaft, at the base of the phallus or scrotum, are less common, accounting for around 35% of cases[3]. Hypospadias is frequently associated with other abnormalities, including abdominal testes (10%) and inguinal hernias (15%), suggesting a common pathology for these male reproductive disorders[4,5].

Of major concern is the unexplained doubling in the incidence of hypospadias in developed countries in recent decades. This is not due to increased reporting and is too rapid to be accounted for by genetic mutations[1,6]. Urethra internalization is tightly regulated by hormones and primarily driven by androgens secreted from the fetal testis. In addition, estrogen also plays a critical role in the masculinization of the penis[7,8]. The process of urethra internalization can be blocked by exogenous estrogen, causing hypospadias in both mice and humans[9,10]. Recent studies have also found an association between increased estrogenic endocrine disrupting chemicals (EDCs) in human fetal blood and the incidence of hypospadias in infants[11]. Despite this direct link, the hormonal targets that control penis development and urethra internalization are not well understood.

The development of the male penile urethra begins early in development, shortly after the septation of the cloaca into the hindgut and urinary tracts[12,13]. In mice, the division of the cloaca and formation of the genital tubercle (GT) occur between embryonic (E) days 10.5 and 13.5[12]. At this stage, the urethra terminates at the base of the developing GT[12]. Urethra internalization proceeds from E14.5 to 17.5[14] until the urethra is internalized along the entire length of the penis, terminating at the distal tip. Development of the GT and urethral closure involves both androgen and estrogen signaling, along with several genetic pathways, including Hedgehog, fibroblast growth factor, bone morphogenetic protein and Wnt signaling[15]. However, little is known about the interactions between EDCs and these gene networks.

Urethral closure is regulated in part by the Eph/Ephrin, bidirectional signaling molecules that mediate many patterning pathways in early embryonic development[16,17]. Within this pathway, there are EphA and EphB receptors as well as ephrinA and ephrinB ligands, which can bind promiscuously to A- and B-receptors. EphB receptors are transmembrane receptor tyrosine kinases that, when bound to Ephrin ligands, signal forward in the Eph-expressing cell via intracellular tyrosine kinase activation. At the same time, a reverse signaling cascade is triggered in the Ephrin-expressing cell, via phosphorylation of its intracellular cytoplasmic tail[18]. Both Ephrins and Eph receptors are membrane-anchored proteins, and signaling only occurs with cell-cell contact[19–21]. *EPHRINB2* (*EfnB2*) loss-of-function mutants show early embryonic lethality in mice. However, mutations which only affect

[1]School of BioSciences, The University of Melbourne, Victoria, VIC, Australia. [2]Department of Genetics, University of Texas M.D. Anderson Cancer Center, Houston, TX, USA. [3]Baylor College of Medicine, Houston, TX, USA. [4]Department of Molecular and Cell Biology and Institute for System Genomics, The University of Connecticut, Storrs, CT, USA. [5]These authors contributed equally: Deidre Mattiske, Pascal Bernard, Paul E. Gradie. ✉e-mail: a.pask@unimelb.edu.au

https://doi.org/10.1038/s42003-025-09322-y                                                                                        **Article**

*EPHRINB2* reverse signaling result in profound cloacal septation defects and hypospadias, indicating it has a pivotal role in urorectal patterning[16].

We have previously described a novel long non-coding RNA (lncRNA) molecule, *Leat1* (long non-coding *EPHRINB2* associated transcript 1), that is essential for female urogenital patterning and development[22]. Here, we describe an essential role for *Leat1* in male urogenital development. *Leat1* associates with EPHRINB2 (either directly or indirectly) in the penis and is required for the normal expression of *EfnB2* during urethra closure. Mice lacking *Leat1* display a complete lack of urethral closure, leading to severe hypospadias. We further demonstrate that *Leat1* expression is impacted downstream of endocrine disruption and hypothesize that *Leat1* could be a potential target of endocrine disruption in the etiology of hypospadias.

## Results

### Leat1 expression in the male genital tubercle
*Leat1* (Long non-coding RNA *EfnB2* Associated Transcript 1, ENSEMBL accession number AK042353), is located approximately 300 kb downstream from the *EfnB2* gene[22]. Amplification of *Leat1* RNA from the E14.5 male genital tubercle identified two isoforms *Leat1a* and *Leat1b* and confirmed that it produces a unidirectional transcript (Fig. 1a and Supplementary Fig 1a, b). Rapid amplification of cDNA Ends (RACE) was used to define the full-length transcripts (Supplementary Fig 1c). We examined the sub-cellular localization of *Leat1* to determine its potential for functional interactions with proteins and/or genomic DNA (Fig. 1a). After 40 cycles using both standard RT-PCR and quantitative RT-PCR (qRT-PCR), *Leat1* was only detected in the cytoplasmic fraction of male cells from the E14.5 GT (Fig. 1a and Supplementary Fig. 1d). *Leat1* levels were below detection limits in both the cytoplasmic and nuclear fractions of the female GT. Expression of *Leat1* was further examined using qRT-PCR in whole GTs during embryonic development to determine its temporal profile. *Leat1* levels peak in males between E13.5 and E14.5 (Fig. 1b), directly following the testosterone surge from the developing testis and directly prior to the initiation of urethra internalization[23]. *Leat1* expression decreased at E15.5 before remaining consistent from E16.5 through to E18.5. *Leat1* levels were consistently lower in females throughout this window of development. We also examined the spatial distribution of *Leat1* using RNAscope and whole mount in situ hybridization. At E13.5, expression was dispersed throughout the distal and ventral aspects of the GT and concentrated in the urethral plate epithelium (Fig. 1c–f). By E16.5 *Leat1* was at the distal tip of the developing GT, in the urethral epithelium and adjacent mesenchyme, and in the urorectal septum (Fig. 1g–j).

Long non-coding RNAs typically show low sequence conservation between species[24], however, VISTA plot analysis revealed extensive conservation of exons 1 and 3 of the *Leat1* transcript across deeply divergent mammalian genomes (Fig. 1k). This unexpected conservation strongly suggests that *Leat1* function is sequence-dependent. Furthermore, *Leat1* showed positional conservation remaining in synteny with *EfnB2* in human, mouse and even in the distantly related marsupials (which last shared a common ancestor with mice and humans 160 million years ago; Fig. 1l), indicating a functional relationship between these genes. In mice, *Leat1* was positioned approximately 325 kb (350 kb in human and 380 kb in wallaby) downstream of the *EfnB2* transcription initiation site.

### Loss of Leat1 causes hypospadias
An isolated hypospadias phenotype (Fig. 2) was identified in mice with a 50 kb genomic deletion that spanned the *Leat1* gene[25,26] (Supplementary Fig. 1e). No other genes are present within the deletion interval. From 82 adult males with the homozygous deletion, $n = 5$ had a mild hypospadias phenotype, $n = 21$ had an intermediate phenotype, and $n = 14$ had a severe hypospadias phenotype. No hypospadias phenotypes have been identified in any of the heterozygous or wild-type littermates ($n > 100$ in both genotypes). We have also observed mothers killing pups at birth that have a more severe phenotype, suggesting that this phenotype may be under-represented in the adult data. Adult mutant mice show an unfused scrotum, with

significantly reduced anogenital distance (AGD) compared to wild type and heterozygotes adult males (Fig. 2i). There was no significant difference in AGD between adult heterozygotes and wild type mice.

Mutants had a functional anus, indicating this to be a discrete penis/scrotal malformation and not an anorectal phenotype. In wild type mice, the penis is completely internalized in its flaccid state and the urethral meatus is located at the distal tip of the glans (Fig. 2a), whereas in homozygous mutants (Fig. 2b, c) the penis was constantly externalized with the meatus located at the base of the glans (Fig. 2b, c). The mutant penis also presented a dorsal foreskin hood, similar to the foreskin phenotype seen in human hypospadias cases[27](Fig. 2b). The hypospadias phenotype is also observed at birth in homozygous mutants (Fig. 2e) with the urethral meatus located at the base of the penis and the urethra and anus in close proximity (reduced AGD). Histological sectioning shows the urethral plate was formed but not closed in the mutant mouse either at birth (Fig. 2g) or at sexual maturity (Fig. 1h).

Since penis development and the process of urethra internalization are androgen regulated, we looked for evidence of androgen deficiency in our mutant mice. Although androgen levels cannot be measured in early developing embryos, we found that all androgen-responsive tissues, including the epididymis, seminal vesicles and testis, developed normally in the mutant adults (Supplementary Fig 2a), with no significant difference in organ weights from that of wild type adults (testis: $p = 0.4210459$, epididymis: $p = 0.2061318$, seminal vesicles: $p = 0.6817668$) or heterozygote adults (testis: $p = 0.9121533$, epididymis: $p = 0.7843484$, seminal vesicles: $p = 0.7437809$) (Supplementary Fig 2b). This indicates that androgen levels were within the normal range during early development. In addition, the testes developed normally with no sign of hypoplasia. Although spermatogenesis appeared normal in the mutant males (Supplementary Fig 2c), they did not produce offspring due to the physical abnormalities of the penis.

### Investigating the relationship between Leat1 and EPHRINB2 function.
The conservation of synteny between *Leat1* and *EfnB2* and its previously reported function in urorectal development[16,28], led us to further investigate their relationship. First, we characterized *EfnB2* during urorectal patterning in wild type embryos (Fig. 3a). *EfnB2* mRNA was localized to the surface epithelium and mesenchyme of the male genital tubercle and was more abundant in the epithelium of the urethra, surrounding mesenchyme, and preputial swellings (Fig. 3a, Transverse). Sagittal sections show *EfnB2* mRNA localized throughout the epithelium of the urogenital sinus, including the urorectal septum (Fig. 3a, Sagittal). These results are consistent with those reported for mRNA localization in the GUDMAP database at E15.5[29]. Next, we compared *EfnB2* mRNA expression in male wild type and mutant embryos by whole mount in situ hybridization (Fig. 3b). In male wild type embryos at E14.5, *EfnB2* expression was observed within the preputial swellings, urethral epithelium and in the surrounding mesenchyme (Fig. 3b, white arrowheads). *EfnB2* transcripts were substantially less abundant in the GT of *Leat1* deficient male embryos but similarly distributed (Fig. 3b, white arrowheads).

To quantify *EfnB2* downregulation, we performed qRT-PCR to measure mRNA levels in the male GT between E12.5 and E18.5 (Fig. 3c). In wild type mice, *EfnB2* expression peaked between E13.5 and E14.5 before decreasing at E15.5, a pattern similar to that observed for *Leat1* expression (See Fig. 1b). Homozygous deletion of *Leat1* resulted in a significant decrease in the expression of *EfnB2* (by ~50%) at all stages from E12.5 through E18.5 (Fig. 3c). Levels of *EfnB2* in the *Leat1* homozygous male GT trends toward levels of EfnB2 in the female GT, but with stage-specific deviations in both directions (Supplementary Fig 3a). Similar to the spatial expression at E14.5, at P0 the spatial distribution of the EPHRINB2 protein was similar in wild type and *Leat1* homozygous male GTs (Supplementary Fig 3b), despite reduced *EfnB2* during embryonic development. *Leat1* heterozygote embryos at E14.5 did not have significantly different expression of *EfnB2* (Supplementary Fig 3c). We next investigated the potential regulatory role of *Leat1* for *EfnB2* expression.

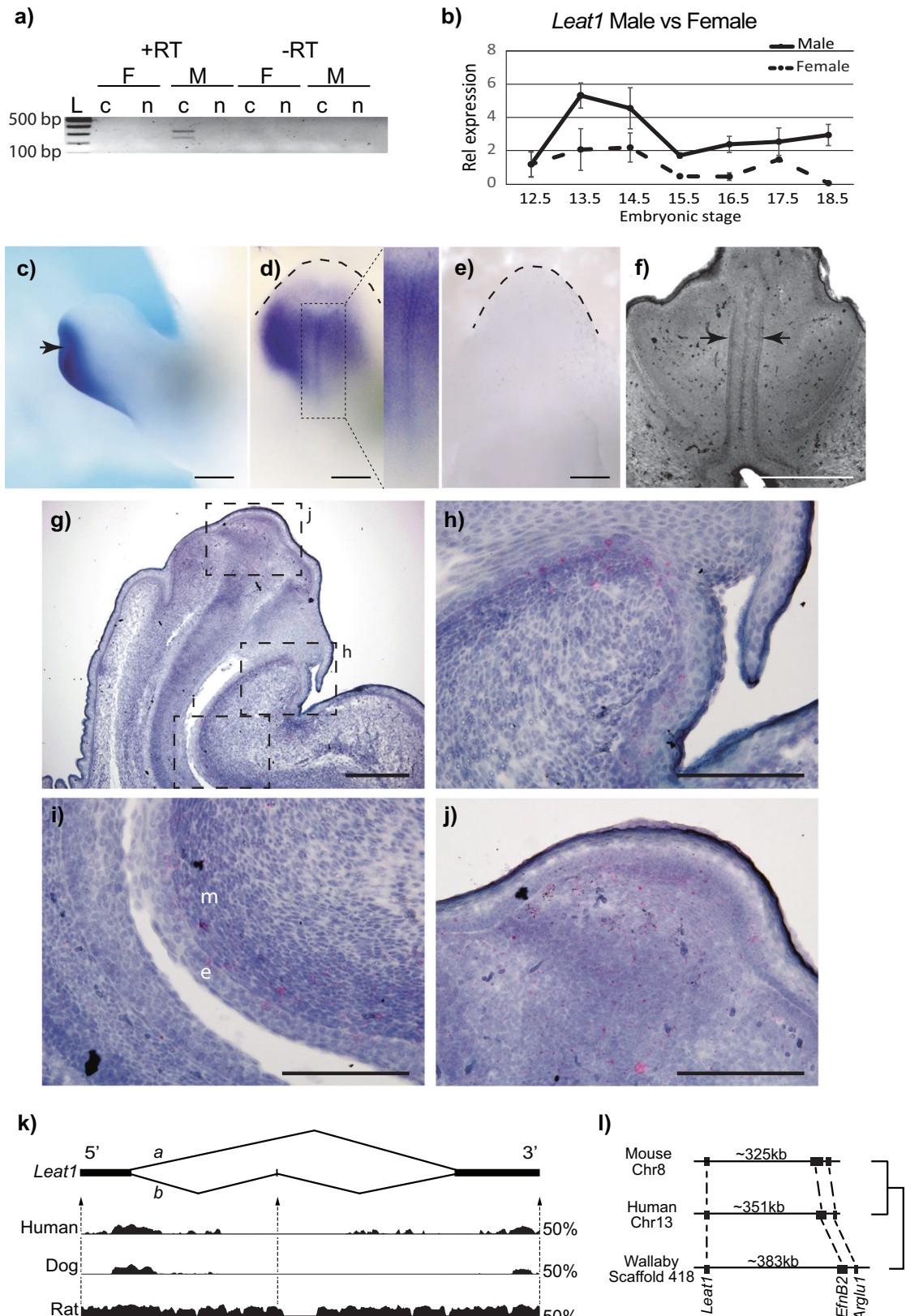

### *EfnB2* expression is regulated by *Leat1*

We used a cell culture system to explore the regulatory relationship between *Leat1* and *EfnB2* expression. The mouse Leydig cell line, TM3[30], has an epithelial phenotype, is responsive to sex hormones and endogenously expresses *EfnB2*. We produced a stably transfected TM3 cell line with an inducible full-length *Leat1a* transgene under the control of a cumate inducible promoter. Expression of *Leat1a* from the transgene in the absence of cumate was low, but still significantly higher than in cells containing empty vector which showed no endogenous *Leat1* expression (Fig. 4a). The expression of *Leat1a* was induced more than 2000-fold by the addition of

**Fig. 1 | Characterization of *Leat1*. a** RT-PCR on RNA extracted from male (M) and female (F) genital tubercles at E14.5. *Leat1* was specifically expressed in male and had a predominantly cytoplasmic localization. The two bands indicate that *Leat1* undergoes alternative splicing. c cytoplasm, n nucleus. L, 1 kb ladder, +RT reverse transcribed template, -RT native RNA template. **b** Quantitative real-time PCR showing *Leat1* expression in the genital tubercle during development in wild type male (blue solid line) and wild type female (dashaed line, *n* = 5). In wild type males, *Leat1* reached maximum expression at E13.5, shortly before the onset of urethra internalization. *Leat1* was consistently lower in females throughout this window. All data presented as mean ± SEM. **c** Lateral view of the GT at E13.5 showing *Leat1* staining (dark blue) in the distal aspect of the developing penis (arrow). **d** Ventral view of the same GT (outlined in dashed line) showing concentrated staining along the urethral plate epithelium (boxed inset), as indicated in (**f**). **e** No staining is seen with the *Leat1* sense probe in the same ventral view of the E13.5 GT (outlined in dashed line) as shown in (**d**). **f** Sagittal section histology showing the shape of the urethral plate at E13.5 (arrows) that is consistent with the *Leat1* in situ staining as seen in (**d**). **g** Sagittal section of the GT at E16.5 showing *Leat1* staining (bright pink) as detected by RNA scope. Boxed insets are shown in higher magnification in (**h, i, j**). **h** *Leat1* is detected (bright pink staining) in the urorectal septum, which migrates distally during urethra closure. **i** *Leat1* is detected in the epithelium and adjacent mesenchyme of the urethra (bright pink staining). **j** *Leat1* staining is detected in both the epithelium and mesenchyme of the distal tip of the GT (bright pink staining). **k** Schematic representation of *Leat1* RNA after sequencing the two products observed in (**d**). *Leat1*a (2 exons) and *Leat1*b (3 exons) 5' and 3' end sequences are shown in Supplementary Fig 1c. VISTA Plots show the conservation of *Leat1* sequence across Human, Dog, and Rat relative to Mouse. Exons 1 and 3 tend to be conserved whereas exon 2 was less so. Y-axis represents 50% identity cutoff. **l** Synteny map showing conservation of relative sequence locations of *Leat1*, *EfnB2*, and Arglu1, *EfnB2*'s closest neighboring gene. Scale bars = 200 μm in (**c–g**), 100 μm in (**h–j**).

**Fig. 2 | Analysis of the hypospadias phenotype in *Leat1* mice. a** Gross morphology of the adult wild type male. The penis sits within the abdominal cavity with the external prepuce visible. **b** Gross morphology of the penis in the mutant mouse line. The penis was externalized and the glans is visible. The prepuce forms a dorsal hood, similar to that seen in human hypospadias. MUMP male urogenital mating protuberance. The penis is lifted in (**c**) to expose the ventral surface. The urethra remains open along the ventral surface of the penis and the urethral opening (meatus) is located at the base of the phallus. There is a deep groove connecting the urethral opening to the anus (indicated with *) and the scrotum is unfused (bifid) in the midline.
**d–g** The mutant phenotype is clearly observable at birth. **d** Gross morphology of a wild type male pup on the day of birth. Red arrow indicates the location of the urethral meatus at the tip of the penis and the blue arrow indicates the position of the anus. The dotted line shows the distance between these two structures (anogenital distance). **e** Gross morphology of the external genitalia in a mutant mouse male pup on the day of birth. The urethral opening is located at the base of the penis (red arrow) and is in close proximity to the anus (blue arrow; dotted lines indicate anogenital distance). **f** Transverse histological section through the middle of the penis on the day of birth stained with hematoxylin and eosin. In the wild type the urethra (U) is centrally located and closed on the ventral surface. In the mutant (**g**) the urethra is open on the ventral surface. **h** Transverse sections through penises from adult wild type, heterozygous and homozygous *Leat1* mice. All sections are taken at the level of the os penis (O). In both wild type and heterozygote males, the urethra is fully enclosed as seen by the lumen (L) surrounded by the corpus cavernosa (CC). In the homozygous *Leat1* mice the urethral lumen remains open and the urethral folds have not fused. **i** AGD was significantly reduced in adult *Leat1* mutants (*n* = 7) compared to both wild type (*n* = 11) and heterozygous (*n* = 9) adult males. ** = *p* < 0.01. Scale bars = 500 μm.

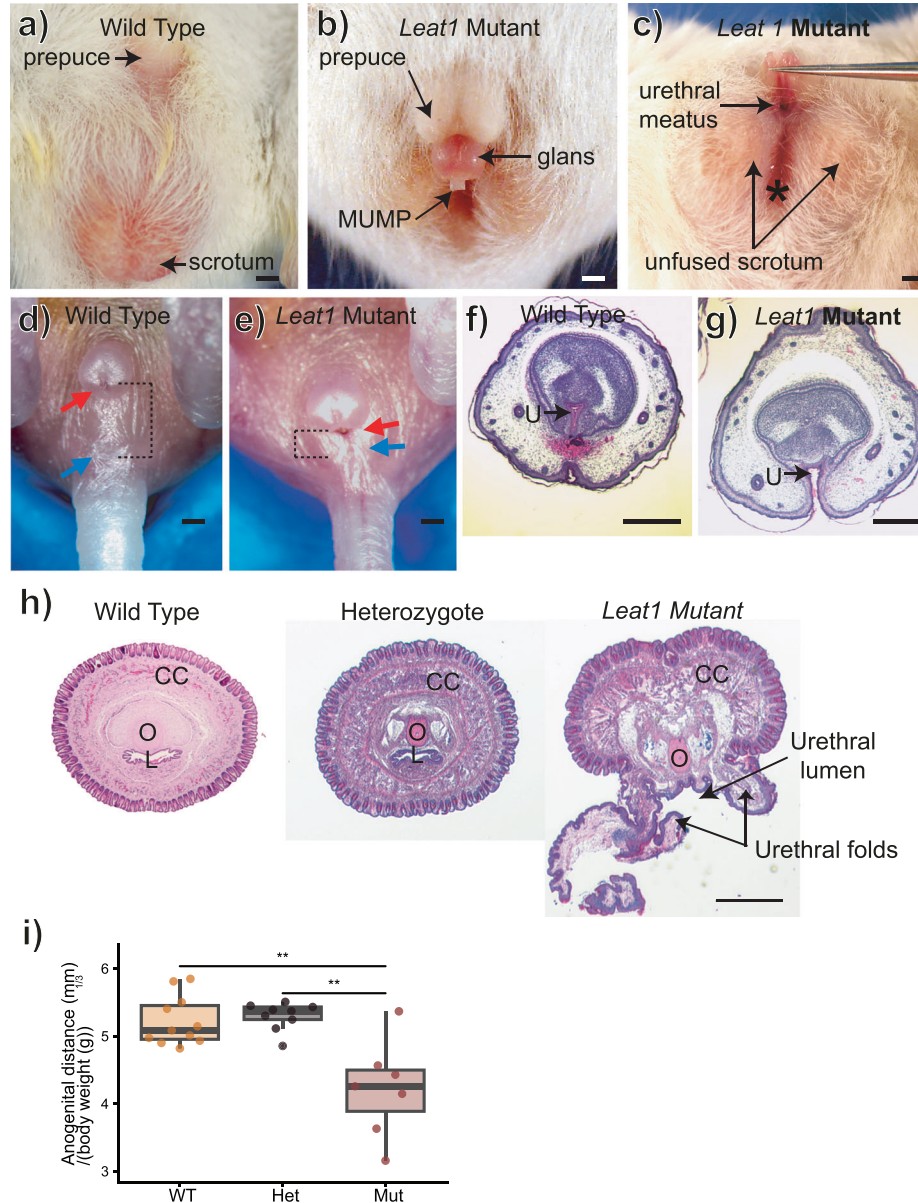

a)

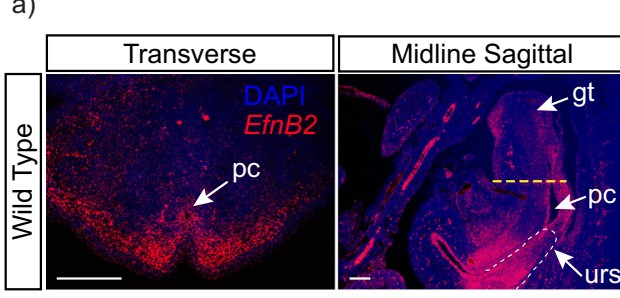

b)

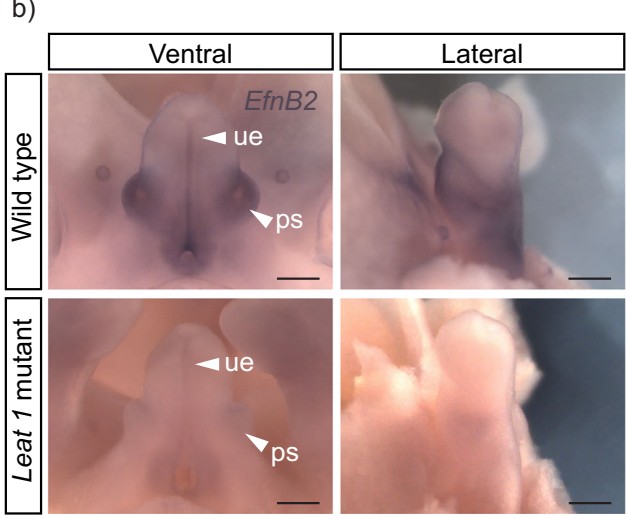

c)

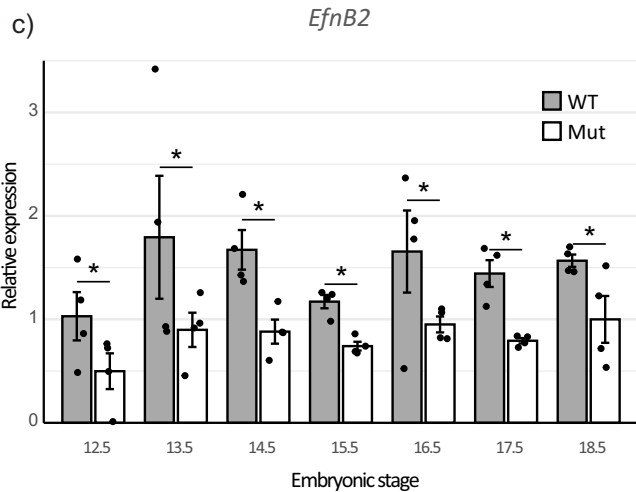

**Fig. 3 | Expression of *EfnB2* in the mouse genital tubercle. a** Expression of *EfnB2* in wild type genital tubercle at E14.5 detected by section fluorescence in situ hybridization showing dense *EfnB2* expression throughout the epithelia and mesenchyme. Yellow dotted line shows approximate section depth of the transverse panel. pc, phallic cloaca; gt, genital tubercle; urs, urorectal septum. Scale bar = 200 μM. **b** Expression of *EfnB2* at E14.5 by whole mount in situ hybridization in wild type and mutant. Ventral and lateral views show strong *EfnB2* expression in the epithelium of the urethra and the preputial swellings (ps). *EfnB2* expression was reduced in the mutant genital tubercle and in the epithelia surrounding the urethra (ue). Scale bars = 200 μm. **c** Quantitative real-time RT-PCR showing relative expression (mean ± SEM) of *EfnB2* in the wild type (black bars) and mutant (white bars) genital tubercle throughout embryonic development (*n* = 4 at each stage). *EfnB2* expression was significantly reduced at all stages by up to 50% in the mutant genital tubercle throughout development. * = *p* < 0.05.

cumate to the culture medium (Fig. 4a). We measured the effect of *Leat1* induction on the expression of endogenous *EfnB2* and observed that *EfnB2* was significantly upregulated (more than 6-fold) when *Leat1* expression was induced (Fig. 4b). Interestingly, even in the absence of cumate, the low level induction of *Leat1* expression was still able to significantly increase *EfnB2* mRNA levels. These results demonstrate that *Leat1* regulates *EfnB2* gene expression at the level of transcription and that it can function in this role in trans, outside of its typical genomic context.

### *Leat1* associates with the EPHRINB2 protein

Given that *Leat1* showed a predominantly cytoplasmic localization, we examined the possibility of it binding to EPHRINB2 protein. Stable TM3 cell lines containing the *Leat1a* inducible construct were transfected with a V5-tagged-*EfnB2* cDNA. After cumate induction of *Leat1a* mRNA, we used the V5 antibody to precipitate the EPHRINB2 protein along with any interacting RNA. RNA was extracted from the precipitate and qRT-PCR was performed to quantify *Leat1a* mRNA. Immunoprecipitation with control IgG did not isolate any V5-tagged protein (Fig. 4c) nor show any amplification of *Leat1a* (Fig. 4d). When the V5 antibody was used to immuno-precipitate protein from cells expressing the V5-tagged EPHRINB2 protein, two bands were detected and (Fig. 4c) we also detected *Leat1a* cDNA with a ~300-fold enrichment over IgG alone (Fig. 4d).

To investigate a physical association in situ between *Leat1* and the EfnB2 protein we performed *Leat1*-EfnB2 proximity ligation assays (PLA) on mouse GT tissue at E17.5. Immunofluorescence of EfnB2 on saggital sections showed staining throughout the urethral epithelium and in mesenchyme of the URS (Fig. 4g, h). PLA foci are present in the urethral epithelium and concentrated in the adjacent penile mesenchyme (Fig. 4i–k), while no foci were detected with *Leat1* sense probe (Fig. 4l). Due to the low abundance of *Leat1*, this is in line with the amount of staining expected with a Leat1 probe.

Proximity ligation assays are highly sensitive for detecting protein–protein interactions in situ, but are limited in that they cannot determine functional interactions or distinguish between direct and indirect associations. Although this doesn't demonstrate a direct interaction, when the PLA and V5-tagged-EfnB2 data are taken together, these data suggest that *Leat1* likely associates with the EPHRINB2 protein, consistent with its predominantly cytoplasmic localization. The cytoplasmic localization of *Leat1* appears to be facilitated by the presence of a 12 bp polyA sequence found at the 3' end of the transcript - as confirmed by rapid amplification of cDNA ends (RACE) – that is present in the genomic DNA (Supplementary Fig 1c). Coincidentally, 12 adenine residues is the minimum required to bind polyA binding proteins[31], to stabilize mRNAs in the cytoplasm.

### *EfnB2* regulates its own expression via a feedback loop involving *Leat1*

EphrinB1 can regulate its own expression[32,33]. To determine whether *Leat1* is involved in the autoregulation of *EfnB2*, we first examined if *EfnB2* can regulate its own expression and second, if this was affected by presence of *Leat1*. We transfected TM3 cells with either the V5 tagged *EfnB2* plasmid or empty vector. Native TM3 cells do not express any endogenous *Leat1*. In the absence of *Leat1*, exogenous *EfnB2* only repressed endogenous *EfnB2* expression by approximately 20% (Fig. 4e). This experiment was then repeated in the presence of the cumate inducible *Leat1a* allele described above. There was a significant suppression of almost 50% mRNA from the endogenous *EfnB2* locus in cells expressing both exogenous *EfnB2* and *Leat1* (Fig. 4f).

### *Leat1* is suppressed by estrogen

We investigated if estrogen could affect *EfnB2* and *Leat1* expression in the developing male GT. GTs were dissected from wild type and *Leat1*-deficient male embryos at E12.5 and cultured for 48 h in hanging drop culture in the presence of dihydrotestosterone to drive virilization (as previously described[34–37]; controls), and with the addition of estrogen (17β-ethinylestradiol; EE2).

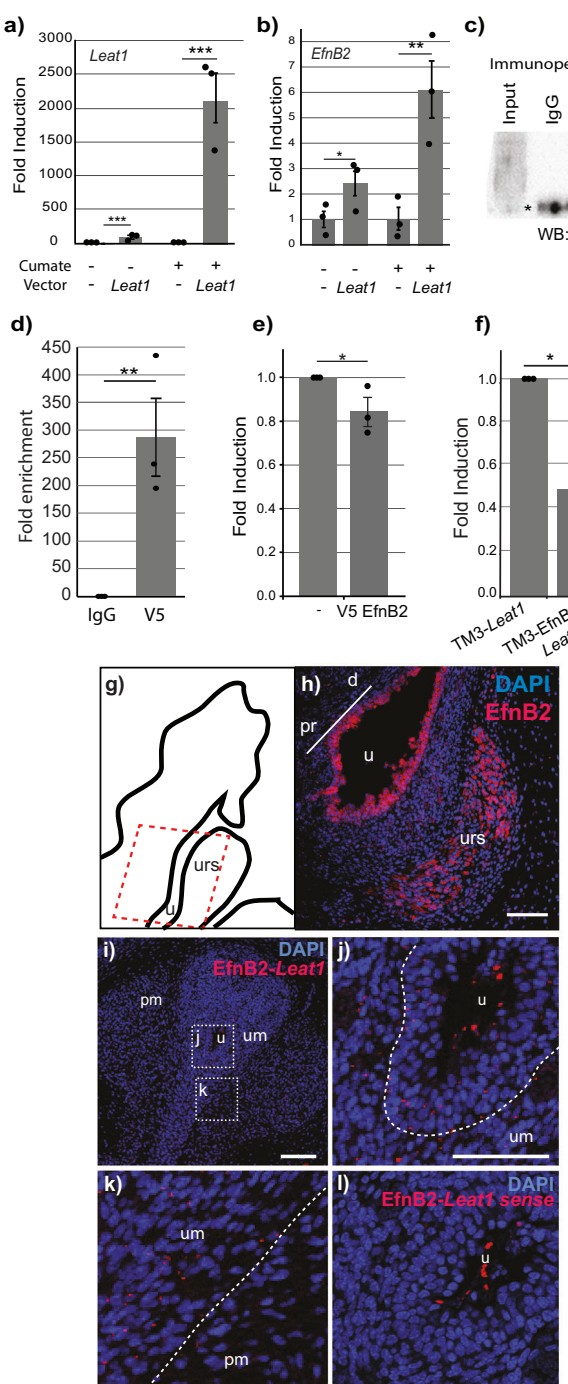

**Fig. 4 | *Leat1* regulates *EfnB2* expression and binds to the EPHRINB2 protein.**
**a** TM3 cells expressing cumate-inducible *Leat1* where treated for 48 h with cumate to induce *Leat1* expression. RNA expression (mean ± SEM) was quantified using real-time quantitative PCR. We saw a more than 2000-fold induction of *Leat1* (black bars, $n = 3$) over non-*Leat1* (gray bars, $n = 3$) expressing cells when *Leat1*-positive cells were treated with cumate. *** = $p < 0.001$. **b** *EfnB2* expression (mean ± SEM) was induced more than six-fold with cumate induced *Leat1* expression ($n = 3$) compared to that of non-*Leat1* expressing cells ($n = 3$). * = $p < 0.05$; ** = $p < 0.01$. **c** Western blot showing V5-tag detection in the immunoprecipitate, indicating efficient pull-down of V5-tagged protein (black arrowheads, alternative splicing) when compared to IgG control precipitation. Asterisk = IgG heavy chain. **d** RNA immunoprecipitation where only protein extracts precipitated using the V5 antibody showed presence of the Leat1 transcript. TM3 cells expressing inducible *Leat1* and V5-tagged *EfnB2* were treated with cumate for 48 h. Proteins were precipitated using the V5 antibody (V5) or control IgG (IgG), RNA extracted and *Leat1* quantified using real-time quantitative RT-PCR (mean ± SEM). Only protein extracts precipitated using the V5 antibody showed presence of the *Leat1* transcript. Fold enrichment was calculated using the ΔΔCt method[58] comparing V5 ($n = 3$) over IgG signals ($n = 3$). ** = $p < 0.01$. **e** TM3 cells which do not express endogenous *Leat1*, were transfected with an empty pcDNA control vector (-) or with pcDNA-V5-*EfnB2*. After 48 h, RNA was extracted and levels of endogenous *EfnB2* quantified by quantitative real-time RT-PCR ($n = 3$). *EfnB2* expression in TM3 cells with a pcDNA-V5-*EfnB2* vector is expressed relative to TM3 cells with an empty vector. * = $p < 0.05$. **f** TM3 cells expressing either inducible *Leat1* alone (TM3-*Leat1*, $n = 3$) or inducible *Leat1* and constitutive *EfnB2* (TM3-*Leat1*/*EfnB2*, $n = 3$) were cultured in the presence of cumate for 48 h. Levels of endogenous *EfnB2* (mean ± SEM) were quantified by quantitative real-time RT-PCR. In the presence of exogenous *EfnB2*, levels of endogenous *EfnB2* were reduced to almost 50%. * = $p < 0.05$. **g** Schematic indicating plane of section (red outline) of g) through the E17.5 mouse genital tubercle. **h** Immunohistology for EphrinB2 in saggital section of E17.5 mouse genital tubercle, showing staining in urethral epithelium and the mesenchyme of the URS (Scale = 200 μm; d distal, pr proximal, u urethra, urs urorectal septum). **i** Proximity-ligation assay demonstrating direct *in vivo Leat1*-EphrinB2 interaction (red foci) in the epithelium of the urethra (u) and adjacent mesenchyme (urethral mesenchyme (um) and penile mesenchyme (pm)) of the E17.5 mouse genital tubercle (average foci per plane section = 536 ± 90). **j** Higher magnification from boxed inset in h) of the foci in the urethra epithelium and adjacent mesenchyme. **k** Higher magnification brom boxed inset in (**h**) of the foci concentrated in the mesenchyme surrounding the urethra (um). **l** No foci were detected using a *Leat1* sense probe in a similar plane to that shown in i) (average foci per plane section = 20 ± 7). Foci within the urethra are non-specific staining (u urethra). Scale bar = 200 μm.

### *LEAT1* is expressed in the human penis

As described above, *LEAT1* is unusually conserved for a lncRNA. Orthologous sequences were readily detected by shared homology across all mammals and always in synteny with *EFNB2*, including in the human genome (Fig. 1k, l). We next determined if *LEAT1* was a functional gene in humans and capable of producing mRNA transcripts in the human penis alongside *EFNB2*[40,41]. We examined *LEAT1* and *EFNB2* mRNA levels in the transcriptomes of urethral plate epithelium (UPE) isolated from patients undergoing repair surgery for mild hypospadias between 6 months and 16 months post-partum (Fig. 6). Although the timing of tissue collection was after the window of urethra internalization in humans, *LEAT1* expression was detected in the UPE of the human patient samples (presented as individual data points in Fig. 6a). Interestingly, *LEAT1* expression levels were variable between samples and very low in 3 out of the 9 hypospadias patients. *EFNB2* was also expressed in the human UPE, as it is in mice, and showed variable levels across patient samples, however, there was no correlation between *LEAT1* and *EFNB2* expression levels (Fig. 6b). These data demonstrate that the human *LEAT1* locus produces an mRNA transcript that is expressed in the urethral plate of the penis alongside *EFNB2*, as in mice.

### Discussion

*Leat1* is a novel, hormonally regulated lncRNA which binds to and facilitates EfnB2 function. Deletion of *Leat1* leads to a suppression in *EfnB2* expression and a complete lack of urethral closure resulting in hypospadias. In addition,

At the end of the culture period, the tissues were snap frozen and gene expression levels examined by qRT-PCR. Connective tissue growth factor (*Ctgf*), a known estrogen responsive gene in the penis[38,39], was used as a control to indicate a positive estrogenic response. *Ctgf* was significantly increased compared to controls in both wild type (Fig. 5a) and *Leat1* deficient GTs (Fig. 5b) cultured with EE2, indicating that the GTs maintained their hormonal responsiveness.

In wild type GTs (Fig. 5a), the addition of EE2 significantly decreased both *Leat1* and *EfnB2* expression. *EfnB2* levels were reduced to around half that seen in wild type embryos, similar to that in the *Leat1* deficient mice and similar to levels seen in the developing female GT (see Supplementary Fig 3a). Significantly, EE2 did not cause a decrease in *EfnB2* expression in the *Leat1* deficient GTs (Fig. 5b).

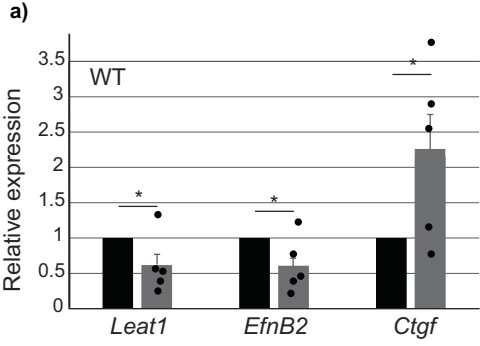

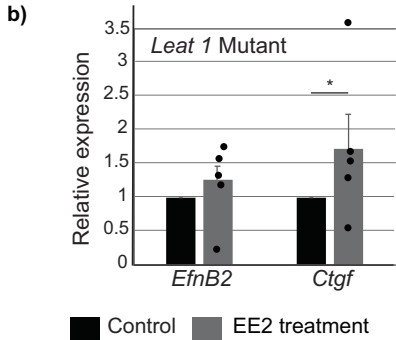

**Fig. 5 | Estrogen regulates *EfnB2* expression via *Leat1*. a** Wild type, male genital tubercles were dissected at E12.5 and cultured over 48 h in the absence (*n* = 5) or presence of 10 nM 17β-ethinylestradiol (EE2)(*n* = 5). After RNA extraction, gene expression was measured by real-time quantitative RT-PCR and presented relative to expression in controls as mean ± SEM. Ctgf was used as a positive control of EE2 action. Both *EfnB2* and *Leat1* expression was significantly downregulated by EE2 treatment, whereas Ctgf expression increased as expected. **b** Mutant, male genital tubercles were dissected at E12.5 and cultured over 48 h in the absence (*n* = 5) or presence of 10 nM 17β-ethinylestradiol (*n* = 5). Expression of *EfnB2* and Ctgf are presented relative to expression in controls as mean ± SEM. In the mutant, in the absence of *Leat1*, *EfnB2* expression was unaffected by EE2 treatment. * = *p* < 0.05, gray bars indicate EE2 exposures, black bars = control.

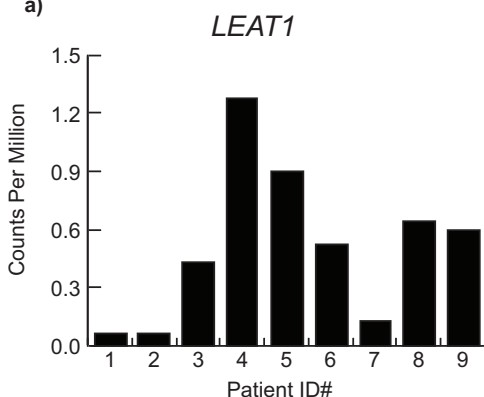

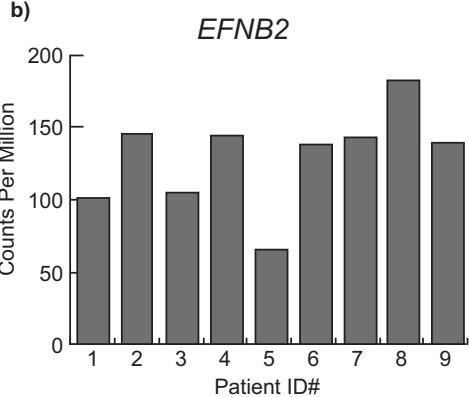

**Fig. 6 | *LEAT1* and *EFNB2* are expressed in the human penis.** Data for both *LEAT1* and *EFNB2* are presented for each corresponding individual patient sample. Reads were aligned to the hg38 Human genome using the Subread aligner and counted using the featureCount function provided in the Subread package. Read counts were normalized using the CPM function from edgeR and expressed as individual data points. Given that the full length of *LEAT1* is unknown in humans, CPM was used in lieu of other length-bias correcting normalization methods. *LEAT1* counts in human were made by extracting reads aligned to the homologous region between mouse *Leat1* and Human chromosome 13 from human RNA-seq UE sample using Samtools. **a** *LEAT1* expression levels in urethral plate epithelium of patients presenting with mild hypospadias. **b** *EFNB2* expression levels in urethral plate epithelium of patients presenting with mild hypospadias.

exogenous estrogen supresses *EfnB2* expression in a *Leat1* dependent manner. Finally, we show that *LEAT1* was conserved in humans and produces a mRNA that was co-expressed alongside *EFNB2* in the human penis. Together, our data identify a new, hormonally regulated driver of urethra internalization that sits at the interface between the genome and the environment in the development of hypospadias.

### *Leat1* associates with and regulates *EfnB2* and is required for urethral closure

We have shown that *Leat1* mRNA physically associates with the EPHRINB2 protein in the developing penis. This is consistent with the predominantly cytoplasmic localization of *Leat1* and its functional poly-A tail. EPHRINB2 sits in the plasma membrane[17] and has a large intracellular tail, which is the only region to which cytoplasmic *Leat1* could bind. To explore how *Leat1*-EPHRINB2 interaction might affect *EfnB2* gene regulation, we examined the impact of this interaction on *EfnB2* autoregulation. In the presence of *Leat1*, exogenous *EfnB2* overexpression was able to suppress transcription of the endogenous *EfnB2* gene by around 50%. However, *EfnB2* transcription was only reduced by approximately 20% in the absence of *Leat1*. Therefore, *Leat1* likely mediates *Efnb2* autoregulation in the developing penis, where both genes are co-expressed.

Loss of *Leat1* in mice caused a complete lack of urethral closure, observable from birth and persisting through to adulthood. The location of the urethral meatus at the base of the penis in homozygous mutant mice, in conjunction with the unfused scrotal bulges, reduced AGD and dorsal

foreskin hood, corresponds to the clinical classification of a severe hypospadias phenotype in humans (Fig. 2b)[42]. The *Leat1* deficient mouse phenotype was remarkably similar to that of heterozygous mice with a mutant form of *EfnB2* (the EfnBlacZ allele[16]). In these mice, the cytoplasmic domain of the EPHRINB2 protein was replaced by a *lacZ* cassette, resulting in a fusion protein lacking reverse signaling[16]. EPHRINB2 reverse signaling is known to regulate epithelial to mesenchymal transitions, a likely critical pathway in mediating the internalization of the urethra[43–45]. *EfnB2* expression was significantly lower in *Leat1* homozygous mutants. Given the defined role of EPHRINB2 in urorectal patterning, the phenotype observed in *Leat1* mutants is consistent with hypomorphic expression of *EfnB2* being the causal factor.

### *Leat1* is a potential target of endocrine disruptors in the etiology of hypospadias

*Leat1* expression was sexually dimorphic and is high in the male GT at E13.5, directly after the onset of androgen production from the fetal Leydig cells[46]. We next investigated the impact of estrogen on *Leat1* and *EfnB2* expression in the developing GT during the window of urethral closure. Male GTs cultured in the presence of DHT showed normal urethral colure and normal upregulation of *Leat1*. In contrast, GTs exposed to estrogen and

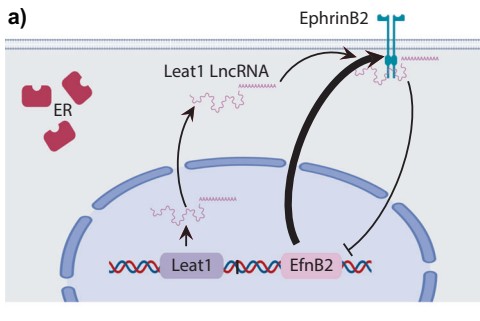

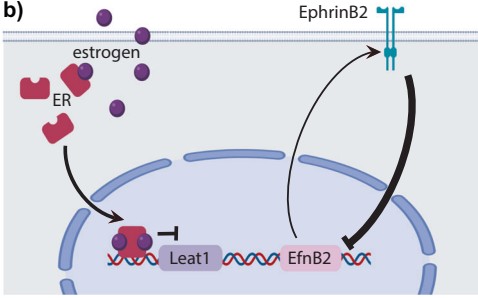

**Fig. 7 | Proposed model of *EfnB2* regulation during mouse urethra formation.
a** *EfnB2* expression is maintained by a feedback loop in which membrane-bound EPHRINB2 signals back to the *EfnB2* gene to regulate its expression. At the onset of urethra internalization, *Leat1* transcripts are upregulated and bind to the EPHRINB2 protein inhibiting the negative feedback loop resulting in elevated *EfnB2* levels. **b** In the absence of *Leat1* (as seen in our mutant mouse line) or following its suppression by estrogen, the EPHRINB2-*EfnB2* negative feedback loop is no longer inhibited. As a result, *EfnB2* levels remain low and urethra internalization is not initiated resulting in hypospadias. Arrows indicate interactions and effects as implicated from our data. Thickness of the arrows indicates the strength of the upregulation (arrow) or inhibition (blunt ended arrow). The precise nature of interactions occurring at the nodes of each arrow is speculative.

DHT showed a significant reduction in the expression levels of both *Leat1* and *EfnB2* to around half of that seen in the normal GT. This was equivalent to *Leat1* and *EfnB2* expression levels in the female GT (Fig. 1b and Supplementary Fig 3a) and an equivalent level of suppression of *EfnB2* to that seen in the *Leat1* null mutant (Fig. 3c). Furthermore, our GT culture system demonstrates that *Leat1* (and *EfnB2*) are directly suppressed by estrogen in the developing penis itself. This suggests EDCs can directly target genes regulating urethral closure in the penis to cause hypospadias, rather than indirectly impacting androgen output from the developing testis[47]. In addition, we showed that the suppression of *EfnB2* by estrogen was *Leat1* dependent. *EfnB2* was not suppressed by estrogen in the developing GTs from *Leat1* null mutants.

Estrogens and estrogen-mimicking endocrine disruptors such as BPA and genistein are well known to increase the risk of hypospadias in mice and humans[11,48,49]. However, the molecular targets of these chemicals are largely unknown. The impact of estrogen on *Leat1* expression implicates it as a direct target of EDCs in the development of hypospadias in human patients. Estrogen exposures do not typically induce severe hypospadias phenotypes, but more mild manifestations of the disease. Exposure to estrogen during later stages of urethra internalization could cause a mild reduction in *Leat1* and subsequently *EfnB2* expression, resulting in mild distal forms of hypospadias. Alternatively, a more potent and early estrogenic exposure during GT development could result in the more severe proximal forms, such as that observed in our model with *Leat1* deficiency.

### *Leat1* conservation suggests a critical function in mammalian genital tubercle development

Long non-coding RNAs are typically poorly conserved at the nucleotide level and only identifiable between species based on their genome location or

secondary structure[50–52]. However, *Leat1* was uncharacteristically conserved across mammals with blocks of high nucleotide conservation between mouse, human and even distantly related marsupial mammals (Supplementary Fig 4). In each case, *Leat1* was located in an identical genomic location syntenic with *EfnB2* (Fig. 1l). Marsupials last shared a common ancestor with mice and humans 160 million years ago[53], indicating an extremely conserved function for *Leat1* in mammals that is both location and sequence-dependent. Given that *Leat1* can bind the EPHRINB2 protein, we suggest that the high degree of nucleotide conservation may be important for mediating this interaction. Despite being syntenic with *EfnB2*, *Leat1* overexpression in trans can still drive *EfnB2* upregulation, indicating that *Leat1* can regulate *EfnB2*, even outside of its genomic context, as described previously for other long non-coding RNAs[54].

We demonstrated that *LEAT1* is a functional gene in humans, producing an mRNA that was expressed in the developing penis alongside *EFNB2*. Interestingly, there was variable *LEAT1* expression in the UPE of humans with mild hypospadias and an almost complete absence in one third of the cases examined (Fig. 6). This, together with the sequence conservation of *LEAT1*, could suggest a conserved interaction with EPHRINB2 to mediate urethra internalization and implicates it as a potential candidate gene in human hypospadias.

### A model for *Leat1* function in urethra internalization

We have shown that *Leat1* likely associates with the EPHRINB2 protein in vitro and hypothesize that *Leat1* affects the EPHRINB2 autoregulatory feedback loop in the developing GT. We propose that *Leat1* reinforces the EPHRINB2 feedback loop, leading to an upregulation of *EfnB2* in the genital tubercle (Fig. 7a). During normal penis development, the upregulation of *Leat1* at E12.5–E13.5 in males, before the onset of urethra internalization, establishes EPHRINB2 autoregulation leading to expression of *EfnB2* mRNAs to the level necessary for urethra internalization (Figs. 1b, 3c, and 7a). A loss of *Leat1* expression reduces EPHRINB2 feedback, downregulating *EfnB2* mRNA expression. This, in turn, prevents urethral closure, resulting in hypospadias. Exposure to exogenous estrogen during this developmental window can reduce *Leat1* expression, leading to inhibition of *EfnB2* mRNA below required levels, resulting in hypospadias (Fig. 7b). Together, these data suggest an essential regulatory mechanism required for normal urethral closure, and one which can be impacted by exogenous estrogen to cause hypospadias in both mice and humans.

## Conclusions

In conclusion, we have identified a novel lncRNA *Leat1* that regulates urethral closure in mice. *Leat1* regulates the function of its neighboring gene *EfnB2* by affecting its autoregulatory feedback, most likely through protein-RNA interactions. *Leat1* shows extraordinary sequence conservation for a lncRNA, indicating conservation of domains we propose mediate its binding to EPHRINB2. We further show that *Leat1* expression was supressed by estrogen and that this mediates the suppression of *EfnB2*. Importantly, we showed that *LEAT1* and *EFNB2* are also co-localized in the developing human penis. Together, our data provide significant new insights into both the developmental mechanisms and molecular regulation of urethra internalization and implicate *Leat1* as a potential target of endocrine disruption in the etiology of hypospadias in mice as well as humans.

## Methods
### Mice
The OVE442 (Leat1 mutation) mouse (*Mus musculus*) line was obtained from P.A. Overbeek laboratory (Baylor College of Medicine, Houston, USA) and was maintained on a FVB/NJ genetic background[22]. Wild type FVB/NJ mice were obtained from Monash University Central Animal Services (Melbourne, Australia). Mouse lines were bred and maintained in the Bioscience 4 animal facility. Mouse embryos were collected from timed matings with noon of the day on which the mating plug was observed designated E0.5 (0.5 days post coitum). Protocols and use of animals conformed to the National Health and Medical Research Council/

Commonwealth Scientific and Industrial Research Organization/Australian Agricultural Council Code of Practice for the Care and Use of Animals for Experimental Purpose (2004) and were approved by the University of Melbourne Committee on Ethics in Animal Experimentation under the reference 1312882.4. We have complied with all relevant ethical regulations for animal use. Adult male tissue was collected at 8 weeks of age, with wet organ weights measured at dissection for testes, epididymides and seminal vesicles. Data of adult organ weights is presented as mean ± standard error of the mean and statistical significance was calculated using a Student's *t* test. Adult testes and penis tissue were fixed in 4% paraformaldehyde at 4 °C overnight, washed in phosphate buffered saline (PBS) and stored in 70% ethanol until processing in paraffin. After processing and embedding, paraffin embedded tissues were serially sectioned with a section thickness of 7 μm. Tissue for section histology was dewaxed in histolene and counter-stained with Lillie-Mayer hematoxylin and eosin according to standard methods[55]. AGD was measured in 8 week old males by holding the base of the tail and measuring the distance from the base of the GT to the center of the anus with a caliper.

## Cell lines
TM3 cells were obtained from ATCC and maintained in culture in DMEM + Glutamax (Life Technologies, Sydney, Australia) supplemented with 10% fetal bovine serum (Life Technologies), antibiotic and antimycotic solution (Life Technologies, Sydney, Australia) at 37 °C in 5% $CO_2$. Stable cell lines were cultured as above, but in media supplemented with the appropriate selective antibiotics.

## Stable cell lines
TM3-PiggyBac *Leat1*a stable cell lines were generated to allow monitoring of gene expression after induction of *Leat1*a expression by treatment with cumate. To produce TM3-PiggyBac *Leat1*a stable cell lines, TM3 cells were seeded in 6-well plates and transfected with 3 μg of PiggyBac-*Leat1*a plasmid together with 1 μg of Super PiggyBac transposase plasmid using Fugene 6 transfection reagent (Promega, Sydney, Australia). 24 h after transfection culture media was supplemented with 4 μg/ml puromycin to select for positive clones. To produce TM3-PiggyBac Control stable cell lines, TM3 cells were seeded in 6-well plates and transfected with 3 μg of empty PiggyBac Cumate Switch inducible vector (SBI, Palo Alto, CA) plasmid together with 1 μg of Super PiggyBac transposase plasmid using Fugene 6 transfection reagent (Promega, Sydney, Australia). Twenty-four hours after transfection culture media was supplemented with 4 μg/ml puromycin (Life Technologies, Sydney, Australia) to select for positive clones.

To produce TM3-PiggyBac *Leat1*a/pc*EfnB2* stable cell lines, a TM3-PiggyBac *Leat1*a cell line was transfected with 1 μg of pcDNA-V5-*EfnB2* using Fugene 6 transfection reagent (Promega, Sydney, Australia). Twenty-four hours after transfection culture media was supplemented with 0.4 μg/ml Geneticin (Life Technologies, Sydney, Australia) to select for positive clones. When indicated cumate treatment was performed for 48 h at a concentration of 30 μg/ml (1×) in culture medium.

## Genome mutation analysis
We mapped the site of the transgene insertion in the mutant mice to chromosome 8 using standard methods[26]. The transgene insertion event caused a 50 kb deletion in a coding-gene deficient region and was located approximately 300 kb downstream from the *EfnB2* gene (Supplementary Fig 1e). We used genomic PCR to verify the transgene insertion boundaries and show that the associated deleted region was specific to our mutant line compared to the background FVB/NJ mice (Fig. 1i). The transgene boundaries were identified using inverse PCR according to published methods[26]. Genomic DNA was extracted and purified using phenol-chlorophorm-isoamyl alcohol DNA extraction. The genomic DNA sequences were amplified with MyTaq™ DNA Polymerase kit (Bioline, Sydney, Australia). Primer pair B1 was used for mapping the 5' end of the transgene insertion boundary and primer pair B2 used for mapping the 3' of the transgene insertion (Supplementary table 1). PCR products were purified using QIAquick PCR Purification kit (Qiagen, Sydney, Australia) and sequenced by the CTP Sanger Sequencing Service in the Department of Pathology, The University of Melbourne.

## Genital tubercle culture
Genital tubercles were dissected from WT ($n = 10$) and *Leat1* mutant ($n = 10$) male embryos at E12.5. GTs were cultured in carbogen using the hanging drop method in BGJb medium (Life Technologies, Sydney, Australia) supplemented with 3 U/ml bovine insulin (Sigma, Sydney, Australia), 0.1 mg/ml L-ascorbic acid (Sigma, Sydney, Australia), 100 U/ml penicillin and 100 U/ml streptomycin (Life Technologies, Sydney, Australia). 10 nM 5α-Androstan-17β-ol-one (DHT; Sigma, Sydney, Australia) and 10 nM 17βethinylestradiol (EE2; Sigma, Sydney, Australia) were dissolved in 100% EtOH. Explants were randomly assigned to be cultured either with 10 nM DHT ($n = 10$) or with 10 nM DHT and 10 nM EE2 ($n = 10$). Previous studies have shown that in similar culture systems, 10 nM DHT is required for the induction of GT elongation, prepuce formation and cellular pro-liferation. Similarly, 10 nM EE2 induces changes in urethral closure without causing excessive growth suppression and altered differentiation evident at dosages higher than 50 nM EE2[37]. After 48 h, gene expression was analyzed by quantitative RT-PCR (see below). Data is presented as mean ± standard error of the mean, and statistical significance was calculated using a Student's *t* test.

## Plasmid construction
*Leat1* cDNA was amplified by PCR from E14.5 mouse genital tubercle cDNA, using primers Clm353F and Clm353R (Supplementary table 1), encompassing 2159 bp of *Leat1* RNA. The PiggyBac-*Leat1*a inducible expression vector was generated by cloning *Leat1*a cDNA into the NheI and NotI restriction site of the multiple cloning sequence of the inducible PiggyBac vector. pcDNA-V5-*EfnB2* was generated using the pcDNA3.1 directional TOPO expression kit (Invitrogen, Sydney, Australia) following the manufacturer's instructions. The *EfnB2* ORF was amplified by PCR from E14.5 mouse genital tubercle cDNA using primers Clm*EfnB2*F and Clm*EfnB2*R and subcloned into pcDNA3.1D/V5-His-TOPO® vector. pGem-*EfnB2* was generated using the pGEM®-T Easy Vector Systems kit (Promega, Sydney, Australia) following manufacturer instructions. A 736 bp fragment of *EfnB2* was amplified from E14.5 mouse genital tubercle cDNA using primers Clim*EfnB2*F and Clim*EfnB2*R and subcloned into pGEM®-T Easy vector. All plasmids were sequenced by the CTP Sanger Sequencing Service in the Department of Pathology, The University of Melbourne.

## Genotyping and DNA sequencing
To determine genotypes of *Leat1* mice, tail tip (fetus and neonates) or ear notches (adults) were used for genomic DNA extraction using DNeasy Blood and Tissue kit (Qiagen, Sydney, Australia). The genomic DNA sequences were amplified with MyTaq™ DNA Polymerase kit (Bioline, Sydney, Australia) using multiplex primers, OVEMutFwd, OVEMutRev, OVEWTFwd and OVEWTRev, detecting wild type and mutant DNA (Supplementary table 1). Amplicons were sequenced by the CTP Sanger Sequencing Service in the Department of Pathology, The University of Melbourne.

## In vitro transcription
pGem-m*EfnB2* plasmid was used as template for PCR amplification using T7 and SP6 primers (Supplementary table 1). The amplicon was purified using the QIAquick PCR Purification kit (Qiagen) and used as a template for transcription and labeling using the DIG RNA labeling Kit (SP6/T7) (Roche, Sydney, Australia). SP6 transcription produced antisense probe, whereas T7 transcription produced sense probe (control).

## Whole mount in situ hybridization
Mouse embryos collected at E14.5 were fixed in 4% paraformaldehyde (PFA) in PBS for 24 h at 4 °C. Whole mount in situ hybridization was

carried out as described previously[56]. Probe signal detection was performed using an anti-digoxigenin antibody from sheep, conjugated with alkaline phosphatase (Roche). Imaging was performed on an Olympus SZX16 microscope equipped with a Nikon DS-Fi2 Digital Sight Camera and using NIS Element software (Nikon).

## Section in situ hybridization

Penises from mice on the day of birth were fixed in 4% PFA overnight, washed twice in PBS and processed for embedding in paraffin wax. 5 µM sections were taken and used for in situ hybridization, performed according to the manufacturer's protocol (Quantigene ViewRNA ISH Tissue kit, Affymetrix). The *EfnB2* probe was synthesized by Invitrogen based on the mRNA NCBI Reference Sequence Gi: 158508443, Ref: NM_010111.5. Mouse embryos collected at E16.5 were fixed in 10% neutral buffered formalin at room temperature for 24 h, washed in PBS and process for embedding in paraffing wax. Sagittal 5 µm sections were taken and used for RNAscope in situ hybridizationm performed according to the manufacturer's protocol (RNAscope 2.5 HD assay (Red), Advanced Cell Diagnostics) with a Mm-Leat1 probe targeting 2-904 of ENSMUST00000208485.2 (Advanced Cell Diagnostics).

## Section histology

Genital tubercles were fixed in 4% PFA overnight and washed in PBS and stored in 70% ethanol until processing in paraffin. After processing and embedding, paraffin embedded tissues were serially sectioned with a section thickness of 7 µm. Tissue for section histology was dewaxed in histolene and counterstained with Lillie–Mayer hematoxylin and eosin according to standard methods[55].

## Tissue proximity ligation assay (PLA)

Tissue PLA was performed following the protocol of[57].with minor modifications. Sections of 5 µm thickness were dewaxed by two 5 min immersions in histolene and 100% ethanol, followed by 5 min immersions in decreasing concentrations of ethanol (90, 70, and 50%) before immersion in distilled $H_2O$ for 5 min. Tissue was heat-treated by immersion in 10 mM sodium citrate buffer with 0.05% tween-20 (pH 6.0), in a waterbath at 95 °C for 30 min. After washing in distilled $H_2O$ for 5 min, tissue was incubated in 0.1 M HCl at 37 °C for 10 min before immersion in distilled $H_2O$ for 5 min. Tissue was subsequently equilibrated with ISH buffer (2× SSC, 0.05% tRNA, 0.2% BSA in PBS) for 5 min at room temperature. 100 nM of Biotinylated *Leat1* probes (Integrated DNA technologies: antisense: GGGAATAAAAG CGGGGACTAGACCTTCTGCCTAAAAATAGTCAAT; sense: TGCTA TCGTGAATCGGATATTAGTCGTTAACGAGACGCTCCTTGA), were applied to tissue and incubated overnight at 37 °C. After incubation with Leat1 probe, tissue was washed twice in 2× SSC and blocked for 30 min with 10% normal donkey serum in PBS. After one 5 min PBS wash, tissue was incubated with anti-EfnB2 (#ab131536, Abcam) and anti-biotin (#ab201341, Abcam) primary antibodies diluted in PBS with 10% normal donkey serum, for 1 h at room temperature, followed by an additional two 5 min PBS washes. Next, plus and minus oligonucleotide-labeled PLA antibodies were applied (#DUO92001 & #DUO92005, Thermo Fisher Scientific), ligation and polymerization performed (#DUO92007, Thermo Fisher Scientific) following the manufacturers instructions. Sections were counterstained with DAPI (100 nM) for 10 min at room temperature. Tissue was imaged on a Nikon A1R confocal microscope. Some non-specific reactivity is present in apical urethral cells and blood vessels in sections that received sense probe, and this is not considered to be part of the PLA signal. Staining was semi-quantified by averaging the number of foci per section for three different sections from three different animals for both sense and antisense probes.

## Nuclear and Cytoplasmic RNA extraction

Nuclear and cytoplasmic RNA were extracted from E14.5 genital tubercles using the Cytoplasmic and Nuclear RNA Purification kit (Norgen Biotek Corp. ON, Canada). Tissues were frozen in liquid nitrogen and ground using mortar and pestle. RNA was extracted following the manufacturer's instructions. RNA was eluted in 40 µl elution buffer and yield was approximately 200 ng/µl for cytoplasmic RNA fraction and 40 ng/µl for nuclear RNA fraction. RNA concentrations were determined using Nano-Drop ND-1000 Spectrophotometer. cDNA was prepared from 200 ng of RNA using Random Primers and the SuperScript III First Strand Synthesis System (Invitrogen, Sydney, Australia). cDNA was amplified with MyTaq™ DNA Polymerase kit (Bioline) using primers, PCR-F and PCR-R, detecting both *Leat1*a and *Leat1*b isoforms.

## RACE analysis of *Leat1* transcript

Rapid amplification of cDNA ends (RACE) was performed using the SMART RACE 5'/3' kit protocol from Invitrogen. Total RNA was extracted from E13.5 genital tubercles and used for RACE amplification following the manufacturer's instructions. PCR products were purified using QIAquick PCR Purification kit (Qiagen, Sydney, Australia) and sequenced by the CTP Sanger Sequencing Service in the Department of Pathology, The University of Melbourne.Sequences were aligned using T-Coffee[58].

## Quantitative real-time RT-PCR

Genital tubercles were dissected from male and female embryos from both wild type and *Leat1* mutants at embryonic stages E12.5 through to E18.5 ($n = 4$ for each stage). RNA extractions were performed using the GenElute Mammalian Total RNA Kit (Sigma, Sydney, Australia). RNA concentrations were determined using NanoDrop ND-1000 Spectrophotometer. cDNA was prepared from 800 ng of total RNA using Random Primers and the SuperScript III First Strand Synthesis System (Invitrogen, Sydney, Australia). All qPCR was performed on a Stratagene MX300P using QuantiTect SYBR Green (Qiagen, Sydney, Australia). *Leat1* primers were designed to amplify exon 1 of the gene, detecting both *Leat1*a and *Leat1*b. Actin and Hprt were used as housekeeping genes for normalization (Supplementary table 1). Relative quantification of gene expression was calculated according to the Pfaffl method[59]. Data were presented as mean ± standard error of the mean and statistical significance was calculated using a Student's *t* test.

## Analyses of *LEAT1* expression in human urethral plate epithelium samples

A small piece of urethral plate epithelial tissue was collected from patients with mild hypospadias phenotypes during surgical repair. Repair surgeries were conducted between 6 months and 16 months post-partum. Samples were collected under ethics application HREC 35189, The Royal Children's Hospital Melbourne. RNA was extracted using the Qiagen RNeasy kit and concentrated using the Qiagen MinElute cleanup kit (Qiagen, Sydney, Australia). Libraries were constructed and sequenced at the Flinders Genomics Facility, Flinders Medical Centre, South Australia, using the TruSeq Stranded mRNA Library Prep and sequenced on an Illumina HiSeq with >160 million reads per sample for the detection of low-expressed transcripts such as lncRNAs, including *LEAT1*.

RNAseq data were assessed for quality using FastQC (http://www.bioinformatics.babraham.ac.uk/projects/fastqc). Reads were aligned to the hg38 Human genome using the Subread aligner. Ten bases were trimmed from both read ends prior to alignment, and only uniquely mapping reads were returned from the alignments. Read counts were performed using featureCount available from the Subread package for R. Only primary alignments with a mapping quality greater than 10 were considered. *LEAT1* counts in human were performed by extracting reads aligning to the *LEAT1* locus on Chromosome 13 using Samtools. The resulting counts were concatenated to the featureCount data prior to normalization. For within-sample expression quantification, counts per million (CPM) were calculated using the edgeR package. The full *LEAT1* transcript length is unknown in humans, so other length correction normalization methods were considered inappropriate. Since these samples were collected from patients aged between 6 and 16 months, all data points are expressed as individual samples.

## RNA immunoprecipitation

RNA protein immunoprecipitation was performed using a Magna RIP kit (Millipore, Sydney, Australia). Briefly, stable TM3 cell lines containing the *Leat1*a inducible construct were transfected with a V5-tagged-*EfnB2* cDNA. Cells were treated with cumate (30 µg/ml) to induce *Leat1* expression. After 48 h, $1 \times 10^7$ cells were washed with ice-cold PBS and lysed on ice with RIP lysis buffer. Protein G magnetic beads (Dynabeads, ThermoFisher Scientific, Sydney, Australia) with antibodies against the V5 tag (ThermoFisher Scientific, Sydney, Australia) or control IgG, were incubated with the cell lysate overnight at 4 °C. Precipitated V5-EPHRINB2 protein or control precipitate were incubated with 10% SDS and proteinase K and the supernatant was used for RNA extraction. After RNA purification and precipitation, RNA was resuspended in 30 µl of RNase-free $H_2O$. Ten microliter of RNA was used for qRT-PCR.

## Immunoprecipitation

Stable TM3 cell lines containing the *Leat1*a inducible construct were transfected with a V5-tagged-*EfnB2* cDNA. Cell were lysed with RIPA buffer (50 mM Tris HCl, pH 8.0, 170 mM NaCl, 0.5% Nonidet P-40, 0.5% Sodium deoxycholate, Complete protease inhibitors). Cell lysates were centrifuged at $14,000 \times g$ for 10 min, and supernatants were used for immunoprecipitation. After preclearing for 1 h with dynabeads protein G (ThermoFisher Scientific, Sydney, Australia), supernatants were incubated at 4 °C overnight with antibodies against V5 tag or control IgG and 50 µg of protein G Dynabeads. Immunoprecipitates were washed five times with lysis buffer, and proteins were eluted with SDS loading buffer, resolved on SDS Page gel and analyzed by immunoblotting using the antibodies as indicated.

## Statistics and reproducibility

Unless otherwise stated, data are presented as mean ± standard error of the mean. Statistical significance was determined using a two-tailed unpaired Student's $t$ test for two-group comparison. Statistical difference was shown as $*P < 0.05$, $**P < 0.01$, and $***P < 0.001$. The detailed information on the sample size and number of replicates in each experiment are given in the respective sections of methods and figure legends.

## Reporting summary

Further information on research design is available in the Nature Portfolio Reporting Summary linked to this article.

## Data availability

A subset of the RNA-sequencing data reported in this study was generated from human-derived samples and is subject to ethical and institutional restrictions. Under the terms of our Human Research Ethics Committee (HREC) approval and participant consent, raw sequencing files cannot be deposited in a public repository because they contain potentially identifiable genomic information. Processed data used in the manuscript (CPM values for all analysed genes) are provided in Supplementary Data. Additional access to the controlled human RNA-seq dataset may be granted to qualified researchers who (i) provide evidence of institutional ethics approval for human genomic data, (ii) agree to the conditions of use specified in our HREC approval, and (iii) sign a data-access agreement ensuring the protection of participant privacy. Requests should be directed to the corresponding author (ajpask@unimelb.edu.au) and will be reviewed in consultation with the University of Melbourne Human Research Ethics Committee. The source data behind the graphs can be found in Supplementary Data.

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

## Acknowledgements

The authors acknowledge the scientific and technical assistance of the Bioscience Electron Microscopy Laboratory at the University of Connecticut. Research reported in this publication was supported by the National Institute of Diabetes and Digestive and Kidney Diseases of the National Institutes of Health under award number R01DK096263.

## Author contributions

D.M.M., P.B., P.E.G., R.J.O., R.R.B. and A.J.P. designed the experiments. D.M.M., P.B., P.E.G., G.T., T.P., R.R.B., N.Y., M.K.S., and A.J.P. performed the experiments. P.A.O. created the mutant mouse line. D.M.M., P.B., P.E.G., and A.J.P. wrote the manuscript. All authors edited the manuscript.

## Competing interests

The authors declare no competing interests.
