## [Transparent Peer review file · Communications Biology]

A long non-coding RNA Leat1 mediates the hormone responsiveness of EfnB2 during male urogenital development

Corresponding Author: Professor Andrew J Pask

Version 0:

Reviewer comments:

Reviewer #1

(Remarks to the Author)

The authors describe a long non-coding RNA Leat1 that is closely linked to the EFNB2 gene and they suggest is able to affect expression and perhaps physically bind to the EphrinB2 protein, to affect male urethral development. While of potential interest and building off of previous studies that indicate EFNB2 is a key player in urogenital development, the current study presented in this manuscript appears premature with too many questions surrounding the experimental design and how well the data supports the author's claims. The possible binding of leat1 RNA to EphrinB2 protein to regulate its function would be exciting, but the study falls short of digging into this more and understanding how/where leat1 may bind and precisely what this possible binding does to EphrinB2 function/activity/reverse signaling/forward signaling, etc.

Not clear if WT controls used in analysis were littermates to the Leat1 embryos?

The studies looking at association of Leat1 RNA with EFNB2 protein are weak and poorly controlled. In 4c, we need to look at both uninduced and induced lysates from both untransfected as well as transfected cells. Such co-immunoprecipitation studies need to be confirmed using embryos for the appropriate stage(s).

Overexpression studies are difficult to assess because so much more EFNB2 mRNA and protein is expressed in the transfected TM3 cells, way more than in a normal cell, and so not sure how massive overexpression of two RNA species might lead to technical artifacts. Again, in vivo studies from embryos are needed.

General presentation of the graphic data is weak, we need to see scatter plots of all the data.

Reviewer #2

(Remarks to the Author)

I read the article with great interest. I found the study to be very interesting and straight forward. By performing all relevant experiments in vitro and in genital tubercle culture, the authors proved that Leat1 regulated EFNB2 gene expression. The only question I have is regarding the association of Leat 1 with EPHRINB2. What was the rationale behind this experiment? Rest everything looks good and I am happy to endorse the publication of this manuscript.

Reviewer #3

(Remarks to the Author)

1. Brief summary of the manuscript

The authors describe an interesting urogenital phenotype in male mutant mice lacking the long non-coding RNA (lncRNA) molecule, leat1. These mice display urethral closure defects that become manifest as hypospadias. leat1 genomic location is 300kb downstream from the EfnB2 gene. Since ephrinB2 is involved in urorectal patterning, leat1 mRNA was detected in the genital tubercle at E12.5-E18.5 and ephrinB2 expression was reduced in mutant mice lacking leat1, a relationship between leat1 and ephrinB2 was investigated. It is suggested that leat1 binds directly to ephrinB2 protein and by this, promotes ephrinB2 expression. Evidence for this shall come from over-expression of an inducible leat1 containing vector in

a cell-culture followed by q-PCR and RNA immunoprecipitation respectively, as well as PLA on tissue sections. Further, estrogen treatment of genital tubercle cultures resulted in a decrease in *leat1* as well as *ephrinB2* expression in wildtype while the level of *ephrinB2* mRNA was not decreased in mutant mice lacking *leat1* which implicates a potential susceptibility of *ephrinB2* to estrogen via *leat1*. *Leat1* and *ephrinB2* mRNA was detected in lysates from the urethral plate epithelium (UPE) isolated from patients with mild hypospadias aged 6-16 months.

The *leat1* mutant phenotype and the indications of a regulatory connection between *leat1* and *ephrinB2* in urogenital development are of potential high interest for the field, the presented results are novel and the conclusions original.

Major issues:

1. A major flaw is that throughout the manuscript, quantification and statistical data are missing, significance is claimed without providing any further quantification information on the respective values and test used. Claims are made and graphs are presented without information about number of samples, technical or biological replicates, whether SEM or SD is shown or often, no error/deviation is shown at all (as in in Fig.6, 4d and Suppl. 3b). At no point, information about the statistical test for significance is provided. Consequently, the often-claimed significance is not supported by evidence. To omit such essential information in a scientific manuscript does not speak for quality and credibility. In addition, several citations are incorrect, in more detail explained under specific comments.

2. Presented evidence to support claims and conclusions is weak:

2.1 Not sufficient evidence for *leat1* being exclusively present in the cytoplasmatic fraction. If the RT-PCR was not sensitive enough to detect *leat1* in female GT, but the qRT-PCR did (albeit in whole GT), this should be repeated on the subcellular fractions to provide evidence that *leat1* was indeed only detected in the cytoplasmatic fraction. This is pivotal for the estimation of its function and interaction potential as also low levels of mRNA can have an effect.

2.2. The hypospadias phenotype in mutant *leat1* mice: For proper assessment of the phenotype, the penetrance should be reported for homozygous, heterozygous and wildtype littermates. No such information is given.

2.3. The claim that androgen-responsive tissues, including the epididymis, seminal vesicles and os penis, developed normally in the mutant adults is not supported sufficiently. As further in the Fig. legend 2 stated: "...the seminal vesicles, epididymis and over all testes size were similar between wild type and mutant..."- No quantification data for the size of the respective structures are provided, nor images of sufficient magnification and quality to visualize and compare the morphology are presented.

2.4. While the *leat1* mutant male mice were not able to reproduce due to the penis malformation, the authors claim that "... spermatogenesis appeared normal..." without presenting evidence.

2.5. *EphrinB2* expression in *leat1* mutant male mice: The image provided is not of sufficient quality to visualize the distribution and appreciate equal distribution of *ephrinB2* in the whole mount genital tubercle in mutants as in wildtypes. In situ hybridization on transversal sections should be presented. To strengthen the claim of a regulatory relationship, *ephrinB2* expression level should also be investigated in *leat1* heterozygous mice.

2.6. "...EPHRINB2 was isolated specifically using the V5 antibody..."

this is not sufficient- Immunoprecipitation with anti-V5 followed by a Western blot detection with anti-V5 antibody does not prove presence of EPHRINB2 protein. Use anti-ephrinB2 antibody for Western blot.

2.7. PLA at E17.5 to prove in vivo binding of *leat1* to the EPHRINB2 protein: PLA can by its nature only show in situ proximity of two partner molecules- in this specific case using three antibodies, the resolution of the assay is approximate 30 nm. Neither in vivo nor binding is supported. Further, since the process of urethral closure is the focus, sagittal sections are not appropriate. Urethral closure occurs at the midline, therefore transverse sections should be used instead. This is highly relevant as claims are made regarding the interaction of *leat1* and EPHRINB2 in just this process and sagittal sections by nature can never show the complete structure. Further, E17.5 is quite late, when *leat1* expression has decreased, possibly to similar levels as in females. Complement with data from earlier stages as E14.5.

2.8. "...Immunofluorescence of *EphrinB2* on sagittal sections showed membrane and cytoplasmic labelling throughout the urethral epithelium and in mesenchyme of the URS..."

The images provided are at low magnification and resolution and do not allow such conclusions. Higher magnification images should be provided, and co-staining with a membrane marker should be used to prove the membrane localization.

2.9. *ephrinB2* expression in TM3 cells transfected with inducible *leat1* vector, resp. V5 tagged *ephrinB2* vector and inducible *leat1*: No evidence for production of EPHRINB2 upon transfection with V5 tagged *ephrinB2* vector is given. To prove the functionality of the vector, qRT-PCR should be done on material from cells transfected with only V5 tagged *ephrinB2* vector, compared to the endogenous level in cells transfected with the empty vector. These shall then be compared with *ephrinB2* levels in cells transfected with inducible *leat1* vector, resp. V5 tagged *ephrinB2* vector and inducible *leat1*.

2.10. *leat1* and *ephrinB2* expression in human samples from patients with mild hypospadias aged 6-16 months: no further conclusions can be drawn from these experiments then the sole presence of the two mRNA species in the same sampled tissue. All other statements are misleading. A cellular co-localization necessary for functional interaction is not supported by RNAseq detection in a tissue lysate. Further, the age of the patients at sampling time was far after the time period when the

urethral closure occurs and does therefore not allow conclusions on an involvement in this process. Expression levels in healthy controls would be needed as comparison.

3. To strengthen the evidence for the claims, the following experiments are required:

Georgas et al, Development 2015 doi: 10.1242/dev.117903, describing with the help of state-of the art technologies (3D volume imaging, fluorescent cell type specific markers and genetic lineage tracing) a revised ontology of urogenital development, including the finding that there are significant and critical differences in urethral development in mice and humans including the. Relate the results to these data- in particular which of the steps in urethral closure and internalization is disturbed and how far these are mouse-specific processes and how this can be translated to humans. This will require a more thorough and detailed histomorphological analysis of transversal sections taken from embryos at successive stages of the urethral closure and internalization.

3.1. To further strengthen the evidence, the following experiments are suggested:

EphrinB2 reverse signaling, which is critical for several midline septation events, among others, the urethral closure. Active reverse signaling includes phosphorylation of conserved tyrosine residues on the cytoplasmic tail of the ephrin-B molecule. Does the presence or absence of Leat1 has an impact on EphrinB2 phosphorylation? Use phospho-specific antibodies for ephrinB in Western blot for example.

Before claiming binding and interaction, computational approaches to predict lncRNA–protein interactions should be employed to demonstrate the theoretical possibility for leat1 to bind and interact with EPHRINB2. Consider sequence, structural information and physicochemical property that suggest or implement any interaction between EPHRINB2 protein and leat1.

Leat1 and EphrinB2 genomic loci should be investigated for ER binding sites.

Specific comments, with recommendations for addressing each comment

1. Line 47-49: "...Recent studies have also demonstrated a direct impact of increased estrogenic endocrine disrupting chemicals (EDCs) in human fetal blood on the incidence of hypospadias in infants ..." This is not correct, the cited study found an association between increased concentrations of EDCs in placenta at the time of birth and cryptorchidism or hypospadias in the infants, but has not "demonstrated a direct impact"
2. Line 59-67: the authors try to give an overview on the Eph/ephrin signaling system and its role in development but neglect the fact that there are EphA and EphB receptors as well as ephrinA and ephrinB ligands which can bind promiscuously to A- and B-receptors. If not specified further, the statement "...a reverse signaling cascade is triggered in the Ephrin-expressing cell, via phosphorylation of its intracellular cytoplasmic tail...", in line 62-63 is wrong as ephrinA ligands are attached to the membrane via a GPI anchor and do not possess a cytoplasmic tail. Be more specific or complete and correct the information
3. Line 72-73 "...We further demonstrate that leat1 is a target of endocrine disruption in the etiology of hypospadias in both mice and humans." Not correct, as the study does not provide evidence for estrogen receptor binding or interaction with leat1, neither in mice nor in humans
4. Line 80-88 Not sufficient evidence for leat1 being exclusively present in the cytoplasmic fraction.
5. If the RT-PCR was not sensitive enough to detect leat1 in female GT, but the qRT-PCR did (albeit in whole GT), this should be repeated on the subcellular fractions to provide evidence that leat1 was indeed only detected in the cytoplasmic fraction. This is pivotal for the estimation of its function and interaction potential.
6. Line 85-86 "...leat1 levels peak in males between E13.5 and E14.5 (Fig 1D), directly following the testosterone surge from the developing testis and directly prior to the initiation of urethra internalization..." Provide evidence- data or citation- for the time frame of the testosterone surge in mice.
7. Fig.1: Inconsistent reporting on the stage of the material used for 1A: E13.5 in results/ E14.5 in Fig. legend
8. Fig.1A: indicate kb in the image
9. Fig.1B: provide information on number of samples, type of replicates, SD/SEM, test for significance The in-situ hybridization should be removed from supplementary and added to Fig.1
10. Supplementary Fig.1 D-G Provide control images using sense probes, also showing the urethral plate as in F, further provide information on the generation of the black and white contrast rendering in F, what technique was used or whether this is a sketch
11. Suppl.Fig.4G: To compare E13.5 GT whole mount with a histological section, use transversal section from E13.5, not E15.5!
12. Line 101-111 Provide statistics on the penetrance of the phenotype in heterozygous and homozygous animals. Provide quantifications of the reduced anogenital distance in OVE442 and control mice
13. Line 107-108: If the dorsal foreskin hood in OVE442 mice is to be compared with the corresponding human phenotype, representative images should be shown or literature cited, Fig2B does not show the human phenotype. Alternatively, exchange identical with a more appropriate term as similar.
14. Line 112-118: The data provided do not support the statement that androgen-responsive tissues resp. androgen level were normal in OVE442 mice: Provide images with sufficient magnification and resolution to enable estimation of morphology of epididymis, seminal vesicles and os penis as well as testes in OVE442 and control mice, provide statistical information and data on overall testes size (Line 764-764) Provide evidence for normal spermatogenesis in OVE442
15. Fig.2 Indicate wildtype and OVE442 in the images, define MUMP in the legend
16. Supp Fig.2: indicate all structures that are named in the text, provide images with sufficient magnification and resolution to be able to estimate the morphology
17. Line 120: there is no evidence provided that leat1 regulates the function of ephrinB2, consider rephrasing
18. Line 123 no data for EPHRINB2 protein expression in Fig.3
19. Line 131-132 the data shown do not allow estimation of whether efnb2 expression in the OVE442 mice has similar

distribution as in wildtype, in situ hybridization as in Fig.3A, on sections of OVE442 mice should be provided and compared with wildtype To strengthen the claim of a regulatory relationship, efnb2 expression level should also be investigated in heterozygous mice

20. Line 136: there is no evidence presented for a significant reduction as no data on significance are provided (statistics, test)

21. Line 137: provide statistical evidence and information about the number of samples, type of replicates, what type of error is displayed in the graph and what statistical test was used

22. Fig.3: add scale bars to the subfigures, replace "mutant" with the actual name of the mutation this makes the content easier to grasp,

23. Fig.3b: all the subfigures are quite low magnification, show zoom in or higher magnification

24. Fig.3c: indicate significance

25. Suppl. Fig.3a: indicate significance

26. Line 142-143: this is wrong! The cited work does not investigate efnb2 expression in the TM3 cell line.

27. Fig.4a, b: provide information on number of samples, type of replicates, SD/SEM, type of test used to test for significance

28. Line 154: in vivo is not correct, in vivo means in the LIVING, there are no live data presented

29. Line 160-161: this is not sufficient- Immunoprecipitation with anti-V5 followed by a Western blot detection with anti-V5 antibody does not prove presence of ephrinB2 protein. Use anti-ephrinB2 antibody for Western blot

30. Fig.4C: the whole blot should be shown since the small excerpt does not convince that lanes were all on one and the same blot, also a protein ladder is missing and the size should be indicated, explain why there are two bands in the lane called V5? This blot does not prove that Ephrin B2 was isolated.

31. Fig4D: again statistics missing, number of replicates, type of replicates, SD/SEM and no statistical test was performed

32. Line 163-164: this is wrong: in vivo interaction can only be proven in a live mouse, the PLA is performed on fixed sections from dead animals

33. Line 164: E17.5 is quite late, when leat1 expression has decreased, possibly to similar levels as in females.

Complement with data from earlier stages as E14.5 Since the process of urethral closure is the focus, sagittal sections are not appropriate. Urethral closure occurs at the midline therefore transverse sections should be used instead. This is highly relevant as claims are made regarding the interaction of leat1 and EPHRINB2 in just this process.

34. Line 165: the magnification of Fig. 4 f,g is far from sufficient to be able to distinguish membrane and cytoplasmatic labelling, provide high magnification high resolution images to support this claim.

35. Fig.4: include the control for PLA here (SupplFig.4)

36. Line 169: see comment on in vivo above;

37. Line 169: a PLA does not prove binding, only close proximity- in the used constellation the distance between both partners would be about 30nm. Give information on how many sections from how many animals have been investigated with PLA. Since there is obviously background binding, provide statistics on number of spots per area resp. section or μm^2 , including n of sections, n of animals and statistics on PLA signal quantification in controls (sense) vs antisense

38. Line179: no evidence is provided for the cytoplasmatic and/or membrane localization of efnb2

39. Line177: "Many Ephrins are known to regulate their own expression" not correct, the reference 30 concerns only ephrinB1.

40. Line 181: Suppl Fig.3B: data do not allow any conclusion without proper quantification, statement on number of replicates, type of replicates, displaying SD/SEM and significance test performed

41. Line 182: no significance can be claimed without statically testing the data

42. Fig.4E: provide number of samples, replicates, type of replicates, SD/SEM and statistical test

43. Line 187: indicate the sex of the embryos

44. Line192/Fig.5 : provide number and sex of GT samples, replicates/ type of replicates, SD/SEM and statistical test

45. Line 196: "EfnB2 levels were reduced to around half that seen in wild type embryos" provide data: avg and SD/SEM, n and statistical test

46. Line 197: false citation; the cited study investigates reverse signaling defective variants of ephrinB2, with full ephrinB2 expression in each of the variants, but signaling impairments to different degrees. Signaling defective mutations should not be mixed up with a knockout deletion of the gene. Especially if forward and reverse signaling have to be considered in this system. A signaling defective ligand will still induce forward signaling upon receptor binding.

47. Line208 provide information about the age of the patients in the text. The presence of ephrinB2 and leat1 in human urethral plate epithelium of patients with hypospadias, long time after urethral closure does not prove a role during this process. No data from healthy controls are provided but would be necessary evaluation of a possible preserved function.

48. Line213: even if ephrinB2 and leat1 are detected in the patient material, only a cellular co-localization would be relevant for the claimed function, which remains to be shown.

49. Fig.6 provide information on number of replicates, SD/SEM and statistical test and age of the patient

Other:

Scale bars are missing in Figs. Supp 1D-G, 2, 4 and Fig.3.

Inconsistencies in the text:

-use the same designation throughout the text, such as OVE442 for the mutant mice

General impression of the manuscript:

The lack of proper statistics combined with weak evidence, overselling of claims and erroneous citations cast doubt on the credibility and seriousness of the study.

Reviewer #4

(Remarks to the Author)

The MS describes on a novel hormonally regulated leat1 RNA which binds and facilitates EfnB2 function. Its deletion leads to a suppression of EfnB2 expression with defective urethral closure, hypospadias. In addition, exogenous estrogen suppresses EfnB2 expression.

The work is valued as reporting a novel RNA with its functions for male type urethra formation. It will be also evaluated not only for researchers on male urethra formation but giving impacts on RNA regulation and tissue fusion models.

Some points are puzzling or at least hard to follow by the current form. I have several points to raise the impacts of MS.

Leat1 and EfnB regulation is the central topic for the MS;

They stated (also in Discussion) as Leat1 binding and facilitating EfnB2 function.

Leat1 binding stabilizes EfnB protein? And it leads to transcriptional autoregulation?

The relationship of leat binding-EfnB and EfnB-autoregulation is puzzling, unclear and should be written clear. They just stated many Ephrins are known to regulate their own expression by one sentence. What is known what is still unclear should be clearly stated.

Leat1 binding to EfnB cytoplasmic region;

Again, what is the background and known examples of RNA binding to EfnB cytoplasmic region and known roles for such regulation? Is this the unique case for leat1? or other examples of RNA binding-mediated auto-regulation?

Masculinization, window and induction/suppressive levels ;

Male type urethral formation is the masculinization process and estrogen action is an additional event.

Estrogenic disturbances is intriguing but careful descriptions are necessary for leat1- mediated masculinization.

Males show rather continuous higher level of expression of leat1 and not showing peaked high level expression shape in the masculinization window (compared with other type of regulators? Nat Rev Uro ; vol 15, 358–368 (2018)) . Even in such leat1 developmental curve, its "functional "peak exists?

Related with such masculinization, leat-mediated EfnB induction is significant but looks not robust particularly in vivo (mutants vs WT; rather low degree of induction is observed throughout embryonic stages). This provokes qe that EfnB can be regulated by additional RNAs?

And leat1 has other additional targets than EfnB and leat1 ko phenotype can not be solely explained by EfnB?

Likewise, the suppressive degree of leat1 and EfnB2 by EE2 is not so prominent.

The above points are the feature of RNA mediated masculinization ?(RNA mediated targets regulation ?)

Conservation among species;

The authors describe on the possible androgen actions of DHT.

They mentioned on the " general conserved model" of Leat1 among species.

and stated the conservation of leat1 extending to marsupials.

Careful descriptions are necessary as they sate the general principle of RNA-EfnB among species;

It is generally known that local production and dependency of DHT varies among species even between mouse-human and Im sure they are experts for such hormonal variation to marsupials.

Version 1:

Reviewer comments:

Reviewer #1

(Remarks to the Author)

The authors failed to address my concerns, my opinion of this weak manuscript has not changed. I am not convinced there is any connection between Leat1 and Efnb2.

Reviewer #3

(Remarks to the Author)

The authors have addressed several of the concerns raised in the initial review. However, the revised manuscript still suffers from instances of overinterpretation of data, where conclusions extend beyond what the data can reliably support. The following issues remain:

Lines 153–154 and Supplementary Figure 3A:

“...Levels of EfnB2 in the Leat1 homozygous male GT were similar to the levels seen during female GT development (Supplementary Fig 3A)...”

“...a) Quantitative real-time RT-PCR showing relative expression of EfnB2 (mean \pm standard error of mean (SEM)) in the male mutant (black bars, n=4 at each stage) and female wild type (white bars, n=4 at each stage) genital tubercle throughout embryonic development. EfnB2 expression was, similar to levels seen in the female GT throughout this window...”

This phrasing oversimplifies the actual data. Supplementary Fig. 3A shows significant differences between female and Leat1 homozygous male GTs at E12.5, E16.5, and E18.5. Therefore, the claim that the expression was “similar throughout this window” is misleading.

The authors should rephrase their statements to more accurately reflect the data.

Lines 174–182:

“Leat1 associates with the EPHRINB2 protein” and related Figure 4 with legend

There are still concerns regarding the interpretation of the immunoprecipitation and Western blot data. The description should reflect the experimental evidence precisely. Specifically, if ephrinB2 protein is not directly detected in the Western blot of the immunoprecipitate obtained with the anti-V5 antibody, it cannot be concluded that ephrinB2 was isolated. As currently shown in Fig. 4C, the Western blot confirms the presence of the V5 tag in the immunoprecipitated material — not the presence of the ephrinB2 protein itself. To substantiate the claim that ephrinB2 was immunoprecipitated, a Western blot with a specific anti-ephrinB2 antibody is necessary. Alternatively, the text should be rephrased to reflect the actual data. For example, in the legend:

“...c) Western blot showing the efficiency of protein precipitation with the V5 antibody. EPHRINB2 was isolated specifically using the V5 antibody..”

This is misleading. A more accurate statement would be:

“Western blot showing V5-tag detection in the immunoprecipitate, indicating efficient pull-down of V5-tagged protein.”

Similarly, the description:

“...d) RNA immunoprecipitation showing Leat1 binding to the EPHRINB2 protein...”

is not supported by the data. What is shown is the presence of Leat1 RNA in the immunoprecipitate, which may indicate an interaction but does not demonstrate binding to ephrinB2 unless the presence of ephrinB2 is directly confirmed.

The following phrasing is correct and should serve as a model:

“..Only protein extracts precipitated using the V5 antibody showed presence of the Leat1 transcript...”

I strongly encourage the authors to rephrase the relevant text and figure legends to accurately represent the experimental evidence.

Lines 190–322:

Despite the statement of the authors that in vivo mechanisms have been removed from the text, claims suggesting in vivo relevance persist throughout the discussion. These statements should be reviewed carefully and either removed or clearly qualified to avoid misrepresentation of the experimental scope.

Line 198:

“...Many Ephrins are known to regulate their own expression^{32,33}...”

This statement remains inaccurate. References 32 and 33 pertain specifically to ephrinB1, and do not provide evidence that “many ephrins” self-regulate. Reference 33 cites reference 32 and does not introduce new data on additional ephrin family members. The sentence should be revised to reflect this.

Lines 201–205:

“...In the absence of Leat1, exogenous EfnB2 was unable to repress endogenous EfnB2 expression (Supplementary Fig 3D). This experiment was then repeated in the presence of the cumate inducible Leat1a allele described above. There was a significant suppression of mRNA from the endogenous EfnB2 locus in cells expressing both exogenous EfnB2 and Leat1 (Fig 4E). ...”

The description of these findings is difficult to follow because the relevant data are split across two different figures. It would greatly improve clarity if the authors could present these data side by side, as the logical argument hinges on their comparison.

Additionally, the extent of repression should be described with greater precision. The reduction of endogenous EfnB2 expression in the absence of Leat1 but presence of exogenous EfnB2 is about 20% — this should be stated clearly. In the presence of Leat1 and exogenous EfnB2, the repression is around 50%, which is more substantial, but still not complete. This distinction should be discussed to avoid the impression of exaggerated impact. The same applies to line 253 in the discussion. I suggest to rather than describing these effects as binary (“repressed” vs. “not repressed”), to characterize the effect more precisely and reflect on the magnitude and biological significance accordingly.

Reviewer #4

(Remarks to the Author)

I read the revised MS and comments.

About the interaction (Leat1 binding to EfnB2 cytoplasmic region), it is the central issue of this MS.

As also discussed and pointed out by other reviewer, they basically stressed " the current study as the 1st to show Leat1 binding to EfnB" which is true but stating with careful descriptions is very necessary (typically the merits and the limitation of PLA).

Judged by their reply to my comments (and judged by the technical limits as also discussed with other reviewers), their sum Fig (Fig 7) should be also modified and "softened" including question marks in it. They stated softened sentences in the revision but the current scheme Fig7 is NOT.

Also about Masculinization, I agree with the importance of estrogenic actions. But importance of developmental-Masculinization as the Male type urethral formation should be written more as initial backgrounds (currently half or one sentence).

Their reply of Cripps paper is intriguing BUT its for EDC and erectile dysfunction.

Version 2:

Reviewer comments:

Reviewer #3

(Remarks to the Author)

The manuscript has improved, but it still contains overstatements and overinterpretation. Correlations and co-expression are still presented as though they imply direct interaction or causality. For publication, the text must report the observations objectively and clearly separate evidence from speculation.

In this review, I indicate when applicable the authors' previous responses, and my latest comment.

Line 72

"..Leat1 interacts with EPHRINB2 in the penis.."

Consider rephrasing to more correctly reflect what the data *suggest*.

Line 74-75

"We further demonstrate that Leat1 is a potential target of endocrine disruption in the etiology of hypospadias in both mice and humans."

Related to comment on line line 304ff (see further down). No demonstration was provided that Leat1 is a potential target of endocrine disruption in the etiology of hypospadias in humans. Consider rephrasing.

Lines 153–154 and Supplementary Figure 3A:

reviewer's comment from 2nd review: "...Levels of EfnB2 in the Leat1 homozygous male GT were similar to the levels seen during female GT development (Supplementary Fig 3A)..."

"...a) Quantitative real-time RT-PCR showing relative expression of EfnB2 (mean \pm standard error of mean (SEM)) in the male mutant (black bars, n=4 at each stage) and female wild type (white bars, n=4 at each stage) genital tubercle throughout embryonic development. EfnB2 expression was, similar to levels seen in the female GT throughout this window..."

This phrasing oversimplifies the actual data. Supplementary Fig. 3A shows significant differences between female and Leat1 homozygous male GTs at E12.5, E16.5, and E18.5. Therefore, the claim that the expression was "similar throughout this window" is misleading.

The authors should rephrase their statements to more accurately reflect the data.

author's response: "...Our intent here was to show that expression in the Leat1 homozygous male GT is significantly reduced compared to the WT male GT (Fig 3) and is more similar to the WT female (as shown in Supp 3A). We did not intend for the statement to be misleading. We have rephrased this to specifically address each time point: "Levels of EfnB2

in the *Leat1* homozygous male GT were similar to the levels seen during female GT development at E13.5, E14.5, E15.5 and E17.5 and significantly reduced compared to female GTs at E12.5.”

The figure legend has also been corrected “*EfnB2* in the *Leat1* homozygous male GT is significantly reduced compared to female WT GTs at E12.5 and is not significantly different at E13.5, E14.5, E15.5 and E17.5. ...”

reviewer's latest comment:

Thank you for the clarification and rephrasing. I understand that the intent was to highlight that expression in *Leat1* homozygous male GTs trends more closely to WT female levels than to WT male levels. However, if there are statistically significant differences between mutant males and WT females at several stages, it is problematic to describe the levels as “similar.” By definition, significance indicates that the expression levels are not the same, so the claim of similarity becomes misleading.

The data in Supplementary Fig. 3A show both reductions (E12.5) and increases (e.g., E16.5, E18.5) compared to WT females. Selectively highlighting only reductions (or time points with no difference) while ignoring significant increases does not accurately reflect the data. A correct description would be that expression in the mutant males shifts toward the female profile but with stage-specific deviations in both directions.

As an alternative to comparing day by day, a more global comparison could be considered—e.g., by looking at the overall expression trajectory, or using an aggregate measure (such as area under the curve). This would provide a clearer and scientifically more robust way of showing that the mutant males follow the female profile, even if they do not match it exactly at each stage.

It would improve transparency to comment briefly on the observed high variability—whether it reflects true biological heterogeneity, developmental timing differences, or is potentially influenced by the relatively small sample size ($n=4$ per group). This would help readers interpret the data more accurately and avoid the impression of selective reporting.

Lines 190–322:

reviewer's comment from 2nd review: Despite the statement of the authors that *in vivo* mechanisms have been removed from the text, claims suggesting *in vivo* relevance persist throughout the discussion. These statements should be reviewed carefully and either removed or clearly qualified to avoid misrepresentation of the experimental scope.

author's response: We have removed reference to *in vivo* throughout the discussion in line with the reviewer's request.

reviewer's latest comment: There is one more unsupported *in vivo* claim, in Results: see Line 192. I may have overlooked this before. Why does it take three rounds for fixing this issue?

Lines 201–205:

reviewer's comment from 2nd review: ...Additionally, the extent of repression should be described with greater precision. The reduction of endogenous *EfnB2* expression in the absence of *Leat1* but presence of exogenous *EfnB2* is about 20% — this should be stated clearly. In the presence of *Leat1* and exogenous *EfnB2*, the repression is around 50%, which is more substantial, but still not complete. This distinction should be discussed to avoid the impression of exaggerated impact. The same applies to line 253 in the discussion. I suggest to rather than describing these effects as binary (“repressed” vs. “not repressed”), to characterize the effect more precisely and reflect on the magnitude and biological significance accordingly.

author's response: We agree with the reviewer on this point. While we had this data in the figure legend, we now include it in the text as well. “In the absence of *Leat1*, exogenous *EfnB2* only repressed endogenous *EfnB2* expression by approximately 20% (Supplementary Fig 3D). This experiment was then repeated in the presence of the cumate inducible *Leat1a* allele described above. There was a significant suppression of almost 50% mRNA from the endogenous *EfnB2* locus in cells expressing both exogenous *EfnB2* and *Leat1* (Fig 4E).” We have also altered the text at line 253 in the discussion, as suggested: “. In the presence of *Leat1*, exogenous *EfnB2* overexpression was able to suppress transcription of the endogenous *EfnB2* gene by around 50%. However, *EfnB2* transcription was only reduced by approximately 20% in the absence of *Leat1*. Therefore, *Leat1* likely mediates *EfnB2* autoregulation in the developing penis where both genes are co-expressed.”

reviewer's latest comment:

It seems that the 20% / 50% reductions are treated by the authors as equivalent in impact to full repression, as they only replaced the binary description with percentages. The manuscript would benefit from a brief reflection on the magnitude of these reductions and their potential biological significance.

Line 183ff

“...To demonstrate a physical interaction *in situ* between *Leat1* and the *EfnB2* protein we performed *Leat1*-*EfnB2* 184 proximity ligation assays (PLA)...”

“...Proximity ligation assays (PLA) are highly sensitive for detecting protein–protein interactions *in situ*, but are limited in that they cannot determine functional interactions or distinguish between direct and indirect associations in protein proximity. However, when the PLA and V5-tagged-*EfnB2* data are taken together, these data suggest that *Leat1* likely associates with the *EPHRINB2* protein both *in vitro* and *in vivo*...”

As I wrote in the first review “...PLA can by its nature only show *in situ* proximity of two partner molecules...” This means close proximity of about 30-40 μ m. Again, the assay measures proximity, not binding nor physical interaction. Please correct the statements. The data may suggest interaction but do not demonstrate it.

Line 301ff:

“...Despite being syntenic with *EfnB2*, *Leat1* overexpression *in trans* can still drive *EfnB2* upregulation, indicating that *Leat1*

can regulate EfnB2, even outside of its genomic context 54...”

It is not clear which statement/speculation the reference 54 should support. Ref 54 is about long non-coding RNAs in development but does not mention LEAT nor EPHRIN. Consider rephrasing so it becomes clear why the reference is used in this place.

Line 304ff:

“...there was variable LEAT1 expression in the UPE of humans with mild hypospadias and an almost complete absence in one third of the cases examined (Fig 6). This, together with the sequence conservation of LEAT1, suggests a conserved interaction with EPHRINB2 to mediate urethra internalization and implicates it as a potential candidate gene in human hypospadias...”

There is an important logical leap: The absence of LEAT1 mRNA in some hypospadias samples does not demonstrate that loss of LEAT1 causes the phenotype. It could be a secondary effect or unrelated. Consider rephrasing.

Reviewer #4

(Remarks to the Author)

As they now replied to my comments (below >>)

.....>>when the PLA and V5-tagged-EfnB2 data are taken together, these data suggest that Leat1 likely associates with the EPHRINB2 protein both in vitro and in vivo

They stated softened sentences in the revision but the current scheme Fig7 is NOT.

.....>>For clarity, we have not included question marks on the figure legend

I would like them to check the entire sentences fitting to the above their own statements.

Referee expertise:

Referee #1: Eph receptors development

Referee #2: urogenital development/non coding RNA

Referee #3: Eph receptors development

Referee #4: urogenital development

Reviewers' comments:

Reviewer #1 (Remarks to the Author):

The authors describe a long non-coding RNA Leat1 that is closely linked to the EFNB2 gene and they suggest is able to affect expression and perhaps physically bind to the EphrinB2 protein, to affect male urethral development. While of potential interest and building off of previous studies that indicate EFNB2 is a key player in urogenital development, the current study presented in this manuscript appears premature with too many questions surrounding the experimental design and how well the data supports the author's claims. The possible binding of leat1 RNA to EphrinB2 protein to regulate its function would be exciting, but the study falls short of digging into this more and understanding how/where leat1 may bind and precisely what this possible binding does to EphrinB2 function/activity/reverse signaling/forward signaling, etc.

Indeed, we agree that understanding the mechanism through which leat1 RNA binds to EphrinB2 warrants further study, however the current manuscript is the first crucial work that uncovers this potential regulatory function. The experiments presented here constitute a large body of work that together demonstrate an association between a long non-coding RNA Leat1 and the EFNB2 gene, together driving urethral closure in a hormone-dependent manner. We have incorporated the valuable reviewer's comments and added multiple pieces of evidence to substantiate the conclusions made in the earlier version of this manuscript. We strongly believe that this body of work as a whole is a significant contribution to the field of urogenital development and our understanding of the development of hypospadias in males.

Not clear if WT controls used in analysis were littermates to the Leat1 embryos?

Yes, where possible, controls, hets and mutants were collected from littermates. We have included this in the methodology where relevant.

The studies looking at association of Leat1 RNA with EFNB2 protein are weak and poorly controlled. In 4c, we need to look at both uninduced and induced lysates from both untransfected as well as transfected cells. Such co-immunoprecipitation studies need to be confirmed using embryos for the appropriate stage(s). Overexpression studies are difficult to assess because so much more EFNB2 mRNA and protein is expressed in the transfected TM3 cells, way more than in a normal cell, and so not sure how massive overexpression of two RNA species might lead to technical artifacts. Again, in vivo studies from embryos are needed.

Reviewer #3 also had queries surrounding these experiments. While some of these experiments are beyond the scope of this paper, and would be better suited for a follow-up mechanistic paper, we have clarified a number of these experiments in our response to reviewer #3.

General presentation of the graphic data is weak, we need to see scatter plots of all the data.

Where appropriate, data is presented as average +/- SE as is standard for our field. Where possible, we have also included scatterplots on histograms as per the reviewer's request. In some instances (as with the human patient data in Fig 6) these are single data points, and we were careful to present these as single data rather than mislead with a grouping of incomparable data points. We have articulated this more clearly in the results and discussion.

Reviewer #2 (Remarks to the Author):

I read the article with great interest. I found the study to be very interesting and straight forward. By performing all relevant experiments in vitro and in genital tubercle culture, the authors proved that Leat1 regulated EFNB2 gene expression. The only question I have is regarding the association of Leat 1 with EPHRINB2. What was the rationale behind this experiment? Rest everything looks good and I am happy to endorse the publication of this manuscript.

We thank the reviewer for their positive response. Initially these experiments were proposed as we had identified Leat1 in close proximity next to Ephrinb2, and the phenotype of the Leat1 and the reverse signalling EphB2 mice were similar. We believe that the discussion around the association of Leat1 with EPHRINB2 is now clearer with all the additional experiments and edits to the manuscript (as detailed in responses to Reviewer #3).

Reviewer #3 (Remarks to the Author):

1. Brief summary of the manuscript

The authors describe an interesting urogenital phenotype in male mutant mice lacking the long non-coding RNA (lncRNA) molecule, *leat1*. These mice display urethral closure defects that become manifest as hypospadias. *leat1* genomic location is 300kb downstream from the *EfnB2* gene. Since *ephrinB2* is involved in urorectal patterning, *leat1* mRNA was detected in the genital tubercle at E12.5-E18.5 and *ephrinB2* expression was reduced in mutant mice lacking *leat1*, a relationship between *leat1* and *ephrinB2* was investigated. It is suggested that *leat1* binds directly to *ephrinB2* protein and by this, promotes *ephrinB2* expression. Evidence for this shall come from over-expression of an inducible *leat1* containing vector in a cell-culture followed by q-PCR and RNA immunoprecipitation respectively, as well as PLA on tissue sections. Further, estrogen treatment of genital tubercle cultures resulted in a decrease in *leat1* as well as *ephrinB2* expression in wildtype while the level of *ephrinB2* mRNA was not decreased in mutant mice lacking *leat1* which implicates a potential susceptibility of *ephrinB2* to estrogen via *leat1*. *Leat1* and *ephrinB2* mRNA was detected in lysates from the urethral plate epithelium (UPE) isolated from patients with mild hypospadias aged 6-16 months.

The *leat1* mutant phenotype and the indications of a regulatory connection between *leat1* and *ephrinB2* in urogenital development are of potential high interest for the field, the presented results are novel and the conclusions original.

Major issues:

1. A major flaw is that throughout the manuscript, quantification and statistical data are missing, significance is claimed without providing any further quantification information on the respective values and test used. Claims are made and graphs are presented without information about number of samples, technical or biological replicates, whether SEM or SD is shown or often, no error/deviation is shown at all (as in in Fig.6, 4d and Suppl. 3b). At no point, information about the statistical test for significance is provided. Consequently, the often-claimed significance is not supported by evidence. To omit such essential information in a scientific manuscript does not speak for quality and credibility. In addition, several citations are incorrect, in more detail explained under specific comments.

Reviewer #3 requested an addition of statistical data such as number of samples, replicates and SED. Much of this statistical data was contained in the methods, but we have added this to the results where relevant to make this clearer. In particular, the human patient data in Fig 6, as referred to by the reviewer, are single data points and we were careful to present these as single data rather than mislead with a grouping of incomparable data points. We have articulated this more clearly in the methods, results and figure legends. The remaining two panels referred to by the reviewer (Fig 4d and Supp 3b) have been corrected as detailed under specific comments below.

2. Presented evidence to support claims and conclusions is weak:

2.1 Not sufficient evidence for *leat1* being exclusively present in the cytoplasmic fraction. If the RT-PCR was not sensitive enough to detect *leat1* in female GT, but the qRT-PCR did (albeit in whole GT), this should be repeated on the subcellular fractions to provide evidence that *leat1* was indeed only detected in the cytoplasmic fraction. This is pivotal for the estimation of its function and interaction potential as also low levels of mRNA can have an effect.

We repeated these experiments using qRT-PCR on cytoplasmic and nuclear fractions. We detected expression at the limits of the assay in the cytoplasmic fraction, but no expression in the nuclear fraction. Since no Ct values were detected in the nuclear fraction, it is inappropriate to use this data to compare the difference between cytoplasmic and nuclear fractions. In a Pfaffl equation the nuclear fraction would be assigned a Ct value of 40, which is NOT true, since no expression was detected. We have included a figure of the Ct values for *Leat1* and *Hprt* from the cytoplasmic and nuclear fractions in Supplementary Fig 1D, but affirm that our original data with RT-PCR is the most appropriate method to assess subcellular expression.

2.2. The hypospadias phenotype in mutant *leat1* mice: For proper assessment of the phenotype, the penetrance should be reported for homozygous, heterozygous and wildtype littermates. No such information is given.

The penetrance of this phenotype has been added for the homozygous, heterozygous and wildtype littermates (lines 108-113), along with an additional histological panel to Figure 2. We have also noted that the severe phenotypes are possibly under-represented as mothers often kill the most severely affected pups at birth. We have also included the statistical data for the AGD across the three genotypes in Fig 2i.

2.3. The claim that androgen-responsive tissues, including the epididymis, seminal vesicles and os penis, developed normally in the mutant adults is not supported sufficiently. As further in the Fig.

legend 2 stated: "...the seminal vesicles, epididymis and over all testes size were similar between wild type and mutant..."- No quantification data for the size of the respective structures are provided, nor images of sufficient magnification and quality to visualize and compare the morphology are presented.

We have repeated this experiment across the three genotypes quantitating weights of the seminal vesicles, epididymis and testes, and included this in Supp Fig 2b. As stated in the original manuscript, there were no significant differences in these weights between wildtype, heterozygous or mutant littermates, therefore, the text has remained unchanged but statistical p given, and the inclusion of the reference to supplemental data. We have increased the magnification of the images in Supp fig 2a to illustrate the overall reproductive tract morphology. Os penis measurements were not included in this data as CT scanning is required for accurate measurements (see Govers et al 2019 <https://doi.org/10.1096/fj.201802586rr>) and the animal, time and monetary costs of confirming no significant phenotype outweighed the benefit.

2.4. While the *leat1* mutant male mice were not able to reproduce due to the penis malformation, the authors claim that "...spermatogenesis appeared normal..." without presenting evidence. We have included H&E sections of mutant testes that demonstrates normal spermatogenesis within the seminiferous tubules and have also included a comparison of the wild type and heterozygote phenotypes (Supp Fig 2c).

2.5. EphrinB2 expression in *leat1* mutant male mice: The image provided is not of sufficient quality to visualize the distribution and appreciate equal distribution of ephrinB2 in the whole mount genital tubercle in mutants as in wildtypes. In situ hybridization on transversal sections should be presented. To strengthen the claim of a regulatory relationship, ephrinB2 expression level should also be investigated in *leat1* heterozygous mice.

We have increased the magnification of these figures to aid in the visualisation of ephrinB2. We have also included the expression of ephrinB2 in *leat1* wild type, heterozygous and homozygous mutant mice at E14.5 (data from QPCR), a key period of genital tubercle development, coinciding with high levels of *leat1* in wild type embryos and hormone responsiveness (Supplementary Fig 3c). Since we have already shown that there is no phenotype in *leat1* heterozygous mice, doing more developmental stages than this was an excessive, unnecessary use of animals. We have also included EPHRINB2 immuno at P0 to show that although the levels are decreased during embryonic development, the spatial distribution of the protein at birth is similar between wild type and *Leat1* mutant males (Supplementary Fig 3b).

2.6. "...EPHRINB2 was isolated specifically using the V5 antibody..."

this is not sufficient- Immunoprecipitation with anti-V5 followed by a Western blot detection with anti-V5 antibody does not prove presence of EPHRINB2 protein. Use anti-ephrinB2 antibody for Western blot.

We did attempt this, with three separate antibodies, but the EPHRINB2 antibodies were unreliable for accurate results with the Western. Instead, we used a fusion protein between EphrinB2 and a V5 tag. Pulling down the V5 tag with a V5 antibody will therefore pull down the fusion protein containing EphrinB2 – this is a standard molecular technique that is widely accepted and has been used in hundreds of publications. For example, see <http://dx.doi.org/10.1016/j.cell.2015.05.028>.

2.7. PLA at E17.5 to prove in vivo binding of *leat1* to the EPHRINB2 protein: PLA can by its nature only

show in situ proximity of two partner molecules- in this specific case using three antibodies, the resolution of the assay is approximate 30 nm. Neither in vivo nor binding is supported. Further, since the process of urethral closure is the focus, sagittal sections are not appropriate. Urethral closure occurs at the midline, therefore transverse sections should be used instead. This is highly relevant as claims are made regarding the interaction of *leat1* and *EPHRINB2* in just this process and sagittal sections by nature can never show the complete structure. Further, E17.5 is quite late, when *leat1* expression has decreased, possibly to similar levels as in females. Complement with data from earlier stages as E14.5.

Over the last decade, PLA has become a standard methodology for studying the proximal interactions between RNA binding proteins (RBPs) and their target mRNAs in cells. As with all methods, it has advantages and disadvantages but is a useful tool for studying protein interactions. While the results presented here do not show definitively that *Leat1* binds to *EPHRINB2*, when viewed alongside the pull downs, they lend significant weight to the hypothesis that there is an interaction between the two molecules. We have softened the language as requested by the reviewer to hypothesize on the potential of this interaction.

The migration of the urorectal septum (URS) is the driver of urethral closure in the mouse (as detailed in a separate manuscript). As such, we used E17.5 sagittal sections as they clearly demonstrate the migration of the URS leading up to urethral closure. At E14.5, before urethral closure, this is much harder to interpret. Sagittal sections also demonstrate the developing urethra and URS mesenchyme along the length of the GT, whereas signalling in transverse sections is dependent on the depth of sectioning and will vary along the length. We have also included a RNAscope for *Leat1* on 16.5 GT sagittal sections (Fig 1) to more clearly show *Leat1* staining in these structures.

2.8. "...Immunofluorescence of EphrinB2 on sagittal sections showed membrane and cytoplasmic labelling throughout the urethral epithelium and in mesenchyme of the URS..."

The images provided are at low magnification and resolution and do not allow such conclusions. Higher magnification images should be provided, and co-staining with a membrane marker should be used to prove the membrane localization.

We have removed reference to the membrane and cytoplasmic labelling, describing only epithelial and mesenchymal staining. We have also included EphrinB2 immunohistochemistry of wild type and mutant genital tubercles at P0, when there is a clear hypospadias phenotype, showing protein localisation in the epithelium of the urethra (Supp Fig 3b). It should be noted that immunohistochemistry is only spatial and will not give quantitative data as the reviewer requested, however we have included the results of the heterozygotes in the quantitative qPCR data at E14.5 which show no significant difference between wild type and heterozygotes.

2.9. ephrinB2 expression in TM3 cells transfected with inducible *leat1* vector, resp. V5 tagged ephrinB2 vector and inducible *leat1*: No evidence for production of *EPHRINB2* upon transfection with V5 tagged ephrinB2 vector is given. To prove the functionality of the vector, qRT-PCR should be done on material from cells transfected with only V5 tagged ephrinB2 vector, compared to the endogenous level in cells transfected with the empty vector. These shall then be compared with ephrinB2 levels in cells transfected with inducible *leat1* vector, resp. V5 tagged ephrinB2 vector and inducible *leat1*.

We validated protein expression from the transfected vector by pulling down the V5 tagged protein on the Western blot as discussed above. This is a highly validated method.

2.10. *leat1* and *ephrinB2* expression in human samples from patients with mild hypospadias aged 6-16 months: no further conclusions can be drawn from these experiments than the sole presence of the two mRNA species in the same sampled tissue. All other statements are misleading. A cellular co-localization necessary for functional interaction is not supported by RNAseq detection in a tissue lysate. Further, the age of the patients at sampling time was far after the time period when the urethral closure occurs and does therefore not allow conclusions on an involvement in this process. Expression levels in healthy controls would be needed as comparison.

Reviewer#3's concerns are that the human data only shows the two mRNA species are present in the same tissue, and that the tissue is collected post-urethral closure in hypospadias patients. It is incredibly rare to be able to collect this tissue – we can only obtain tissue at the time of urethral repair, and the aim of this experiment was simply to confirm that both *Leat1* and *EfnB2* are present in the human urethra. While we cannot collect this tissue from healthy controls (as suggested by the reviewer), we do believe that this is still important data. Indeed, we have been careful to not overstate these findings and have not suggested a cellular co-localisation, only stating that the two mRNA are found in the samples tissue - see line 233-235 "These data demonstrate that the human *LEAT1* locus produces an mRNA transcript that is expressed in the urethral plate of the penis alongside *EFNB2*, as in mice." We strongly believe that this data should be included, but we would be willing to withdraw this data from the manuscript if it's still perceived to be problematic.

3. To strengthen the evidence for the claims, the following experiments are required: Georgas et al, *Development* 2015 doi: 10.1242/dev.117903, describing with the help of state-of the art technologies (3D volume imaging, fluorescent cell type specific markers and genetic lineage tracing) a revised ontology of urogenital development, including the finding that there are significant and critical differences in urethral development in mice and humans including the. Relate the results to these data- in particular which of the steps in urethral closure and internalization is disturbed and how far these are mouse-specific processes and how this can be translated to humans. This will require a more thorough and detailed histomorphological analysis of transversal sections taken from embryos at successive stages of the urethral closure and internalization.

We agree that this is required in the field, but unfortunately this is beyond the scope of this manuscript and indeed, forms a separate manuscript that is in preparation.

3.1. To further strengthen the evidence, the following experiments are suggested: *EphrinB2* reverse signaling, which is critical for several midline septation events, among others, the urethral closure. Active reverse signaling includes phosphorylation of conserved tyrosine residues on the cytoplasmic tail of the *ephrin-B* molecule. Does the presence or absence of *Leat1* has an impact on *EphrinB2* phosphorylation? Use phospho-specific antibodies for *ephrinB* in Western blot for example.

We did consider this, but the commercial antibodies available for phosphorylation studies are human specific and cross react with other *Ephrin* family members, complicating any interpretation that could be made from these experiments. Further, analysing the phosphorylation of *EphrinB2* wouldn't change the conclusions of this paper. While these experiments would provide some detail into the mechanism of action of *Leat1*, these are significant experiments that are beyond the scope of this paper.

Before claiming binding and interaction, computational approaches to predict lncRNA–protein interactions should be employed to demonstrate the theoretical possibility for *leat1* to bind and interact with EPHRINB2. Consider sequence, structural information and physicochemical property that suggest or implement any interaction between EPHRINB2 protein and *leat1*. *Leat1* and EphrinB2 genomic loci should be investigated for ER binding sites.

Again, we have considered this, but these are considerable experiments and are also fraught with difficult interpretation and error. In the manuscript we decided to address this question directly by looking for a physical interaction between *Leat1* and Ephrinb2 and have done this in the most comprehensive way we can. We demonstrated through pulldown the association of EphrinB2 and *leat1*, their overlapping expression profiles and proximity. Together, these experiments demonstrate that *leat1* either directly binds to or is in a complex with ephrinb2. The reviewer raises a valid point that there may be other components mediating the interaction of *leat1* and EphrinB2 so we have amended the language in the manuscript to reflect this, however this does not change the conclusions of the manuscript.

Specific comments, with recommendations for addressing each comment

1. Line 47-49: “...Recent studies have also demonstrated a direct impact of increased estrogenic endocrine disrupting chemicals (EDCs) in human fetal blood on the incidence of hypospadias in infants ...” This is not correct, the cited study found an association between increased concentrations of EDCs in placenta at the time of birth and cryptorchidism or hypospadias in the infants, but has not “demonstrated a direct impact”

This has been corrected to “Recent studies have also found an association between increased estrogenic endocrine disrupting chemicals (EDCs) in human fetal blood and the incidence of hypospadias in infants”.

2. Line 59-67: the authors try to give an overview on the Eph/ephrin signaling system and its role in development but neglect the fact that there are EphA and EphB receptors as well as ephrinA and ephrinB ligands which can bind promiscuously to A- and B-receptors. If not specified further, the statement “...a reverse signaling cascade is triggered in the Ephrin-expressing cell, via phosphorylation of its intracellular cytoplasmic tail...”, in line 62-63 is wrong as ephrinA ligands are attached to the membrane via a GPI anchor and do not possess a cytoplasmic tail. Be more specific or complete and correct the information

We have rewritten this to be more specific.

3. Line 72-73 “...We further demonstrate that *leat1* is a **potential** target of endocrine disruption in the etiology of hypospadias in both mice and humans.” Not correct, as the study does not provide evidence for estrogen receptor binding or interaction with *leat1*, neither in mice nor in humans

We have softened the language here to “We further demonstrate that *Leat1* is a potential target of endocrine disruption in the etiology of hypospadias in both mice and humans”.

4. Line 80-88 Not sufficient evidence for *leat1* being exclusively present in the cytoplasmic fraction.

As addressed above in 2.1.

5. If the RT-PCR was not sensitive enough to detect *leat1* in female GT, but the qRT-PCR did (albeit in

whole GT), this should be repeated on the subcellular fractions to provide evidence that *leat1* was indeed only detected in the cytoplasmatic fraction. This is pivotal for the estimation of its function and interaction potential.

As detailed above in 2.1.

6. Line 85-86 "...*leat1* levels peak in males between E13.5 and E14.5 (Fig 1D), directly following the initiation of testosterone production in the developing testis and directly prior to the initiation of urethra internalization..." Provide evidence- data or citation- for the time frame of the testosterone surge in mice.

Citation has been added.

7. Fig.1: Inconsistent reporting on the stage of the material used for 1A: E13.5 in results/ E14.5 in Fig. legend

Corrected to 14.5 in results.

8. Fig.1A: indicate kb in the image

Ladder has been added to the image.

9. Fig.1B: provide information on number of samples, type of replicates, SD/SEM, test for significance
The in-situ hybridization should be removed from supplementary and added to Fig.1

Statistical information has been added to the figure legend. The in situ hybridization has been added to Figure 1. We have also complemented this with HCR data showing expression of *Leat1* at a later stage, during URS migration.

10. Supplementary Fig.1 D-G Provide control images using sense probes, also showing the urethral plate as in F, further provide information on the generation of the black and white contrast rendering in F, what technique was used or whether this is a sketch

We have moved these images to Fig 1 and added the sense probe. The black and white contrast was simply a rendering to aid interpretation of the staining in panel F. We have removed this rendering from the figure.

11. Suppl.Fig.4G: To compare E13.5 GT whole mount with a histological section, use transversal section from E13.5, not E 15.5!

We presume that the reviewer meant Suppl Fig 1G, not 4G. The reference to 15.3 was simply a grammatical error, it now reads 13.5.

12. Line 101-111 Provide statistics on the penetrance of the phenotype in heterozygous and homozygous animals. Provide quantifications of the reduced anogenital distance in OVE442 and control mice

We have provided data for the reduced AGD in WT, Het and *Leat1* mutant mice, along with the penetrance of the phenotype. We have included a new plot of AGD (Fig 2i).

13. Line 107-108: If the dorsal foreskin hood in OVE442 mice is to be compared with the corresponding human phenotype, representative images should be shown or literature cited, Fig2B does not show the human phenotype. Alternatively, exchange identical with a more appropriate term as similar.

We have change the term to similar as suggested but also included a reference with the dorsal hood phenotype.

14. Line 112-118: The data provided do not support the statement that androgen-responsive tissues resp. androgen level were normal in OVE442 mice: Provide images with sufficient magnification and resolution to enable estimation of morphology of epididymis, seminal vesicles and os penis as well as testes in OVE442 and control mice, provide statistical information and data on over all testes size (Line 764-764) Provide evidence for normal spermatogenesis in OVE442

As detailed above, we have repeated this experiment quantitating weights of the seminal vesicles, epididymis and testes, and included this in Supp fig 2b. As stated in the original manuscript, there were no significant differences in these weights between wildtype, heterozygous or mutant littermates, therefore, the text has remained unchanged but statistical p given, and the inclusion of the reference to supplemental data. We have also included images of normal spermatogenesis in H&E sections through seminiferous tubules of WT and mutant testes (Supp 2c).

15. Fig.2 Indicate wildtype and OVE442 in the images, define MUMP in the legend

Wildtype and mutant have been added to the images and MUMP defined in the figure legend.

16. Supp Fig.2: indicate all structures that are named in the text, provide images with sufficient magnification and resolution to be able to estimate the morphology

We have increased the magnification and indicated epididymides. We have also included weights of testis, epididymides and seminal vesicles.

17. Line 120: there is no evidence provided that *leat1* regulates the function of *ephrinB2*, consider rephrasing

This has been rephrased to:

*Investigating the relationship between *Leat1* and *EPHRINB2* function*

18. Line 123 no data for *EPHRINB2* protein expression in Fig.3

This has been corrected to only refer to the gene, not the protein.

19. Line 131-132 the data shown do not allow estimation of whether *efnb2* expression in the OVE442 mice has similar distribution as in wildtype, in situ hybridization as in Fig.3A, on sections of OVE442 mice should be provided and compared with wildtype To strengthen the claim of a regulatory relationship, *efnb2* expression level should also be investigated in heterozygous mice

As previously mentioned (see note 2.5).

20. Line 136: there is no evidence presented for a significant reduction as no data on significance are provided (statistics, test)

We have included the statistical data in the figure legend.

21. Line 137: provide statistical evidence and information about the number of samples, type of replicates, what type of error is displayed in the graph and what statistical test was used

Much of this statistical information was already in the methods but we have elaborated where appropriate and reiterated this in the figure legends.

22. Fig.3: add scale bars to the subfigures, replace “mutant” with the actual name of the mutation this makes the content easier to grasp,

Corrected

23. Fig.3b: all the subfigures are quite low magnification, show zoom in or higher magnification

We have included a higher magnification of Fig 3b.

24. Fig.3c: indicate significance

As previously mentioned, this has been added to the figure legend.

25. Suppl. Fig.3a: indicate significance

As previously mentioned, this has been added to the figure legend.

26. Line 142-143: this is wrong! The cited work does not investigate efnb2 expression in the TM3 cell line.

This is simply that the ref is included at the end of the sentence instead of after the statement that it is a mouse Leydig cell line. We have moved the reference appropriately.

27. Fig.4a, b: provide information on number of samples, type of replicates, SD/SEM, type of test used to test for significance

Apologies that this was not clearer. This data has been repeated both in the methods and in the figure legend as appropriate.

28. Line 154: *in vivo* is not correct, *in vivo* means in the LIVING, there are no live data presented

This has been corrected to *in situ*.

29. Line 160-161: this is not sufficient- Immunoprecipitation with anti-V5 followed by a Western blot detection with anti-V5 antibody does not prove presence of ephrinB2 protein. Use anti-ephrinB2 antibody for Western blot

As detailed above, we tried three different EphrinB2 antibodies for our protein experiments. Unfortunately, none of these were reliable enough to use in these protocols. Instead, we validated protein expression from the transfected vector by pulling down the V5 tagged protein on the Western blot as discussed above. This is a highly validated, classic method.

30. Fig.4C: the whole blot should be shown since the small excerpt does not convince that lanes were all on one and the same blot, also a protein ladder is missing and the size should be indicated, explain why there are two bands in the lane called V5? This blot does not prove that Ephrin B2 was isolated.

As this is a Western blot using the V5 antibody, this will not bind to the ladder. The two bands are due to alternative splicing.

31. Fig4D: again statistics missing, number of replicates, type of replicates, SD/SEM and no statistical test was performed

Corrected.

32. Line 163-164: this is wrong: *in vivo* interaction can only be proven in a live mouse, the PLA is performed on fixed sections from dead animals

As above. See point 28.

33. Line 164: E17.5 is quite late, when *leat1* expression has decreased, possibly to similar levels as in females. Complement with data from earlier stages as E14.5 Since the process of urethral closure is the focus, sagittal sections are not appropriate. Urethral closure occurs at the midline therefore

transverse sections should be used instead. This is highly relevant as claims are made regarding the interaction of *leat1* and *EPHRINB2* in just this process.

The migration of the urorectal septum (URS) is the driver of urethral closure in the mouse (as detailed in a separate manuscript). As such, we used E17.5 sagittal sections as they clearly demonstrate the migration of the URS leading up to urethral closure. At E14.5, before urethral closure, this is much harder to interpret. Sagittal sections also demonstrate the developing urethra and URS mesenchyme along the length of the GT, whereas signalling in transverse sections is dependent on the depth of sectioning and will vary along the length.

34. Line 165: the magnification of Fig. 4 f,g is far from sufficient to be able to distinguish membrane and cytoplasmic labelling, provide high magnification high resolution images to support this claim.

As previously mentioned

35. Fig.4: include the control for PLA here (SupplFig.4)

Corrected.

36. Line 169: see comment on in vivo above;

As above.

37. Line 169: a PLA does not prove binding, only close proximity- in the used constellation the distance between both partners would be about 30nm. Give information on how many sections from how many animals have been investigated with PLA. Since there is obviously background binding, provide statistics on number of spots per area resp. section or μm^2 , including n of sections, n of animals and statistics on PLA signal quantification in controls (sense) vs antisense

We have provided the statistical data as requested in both the figure legends and the methods.

38. Line179: no evidence is provided for the cytoplasmic and/or membrane localization of *efnb2*

As above

39. Line177: "Many Ephrins are known to regulate their own expression" not correct, the reference 30 concerns only *ephrinB1*.

The reference of Arvanitis and Davy 2012 Cell Adh Migr. has been added.

40. Line 181: Suppl Fig.3B: data do not allow any conclusion without proper quantification, statement on number of replicates, type of replicates, displaying SD/SEM and significance test performed

Added.

41. Line 182: no significance can be claimed without statically testing the data

Same as 181.

42. Fig.4E: provide number of samples, replicates, type of replicates, SD/SEM and statistical test

Added.

43. Line 187: indicate the sex of the embryos

This has been added into the results, although it was already in methods

44. Line192/Fig.5 : provide number and sex of GT samples, replicates/ type of replicates, SD/SEM and statistical test

This was originally in the methods, but we have added it to the figure legend as well.

45. Line 196: "EfnB2 levels were reduced to around half that seen in wild type embryos" provide data: avg and SD/SEM, n and statistical test

As above.

46. Line 197: false citation; the cited study investigates reverse signaling defective variants of ephrinB2, with full ephrinB2 expression in each of the variants, but signaling impairments to different degrees. Signaling defective mutations should not be mixed up with a knockout deletion of the gene. Especially if forward and reverse signaling have to be considered in this system. A signaling defective ligand will still induce forward signaling upon receptor binding.

The reference has been removed from here and left for the discussion.

47. Line 208 provide information about the age of the patients in the text. The presence of ephrinB2 and *Leat1* in human urethral plate epithelium of patients with hypospadias, long time after urethral closure does not prove a role during this process. No data from healthy controls are provided but would be necessary evaluation of a possible preserved function.

We have included the age of the patients in the text. Unfortunately, since this tissue is only collected during repair surgery, there is no equivalent control tissue that can be obtained. Due to this limitation we are careful not to overreach on this significance, but we do believe it is important that both *EFNB2* and *Leat1* are found in this tissue. We have been careful not to overstate this in either the results or the discussion. As stated above, we would be willing to exclude this from the paper, but do feel strongly that it should be included.

48. Line 213: even if ephrinB2 and *Leat1* are detected in the patient material, only a cellular co-localization would be relevant for the claimed function, which remains to be shown.

As above, we have been careful to articulate this clearly, and do not claim anything beyond the presence of both in the same tissue.

49. Fig.6 provide information on number of replicates, SD/SEM and statistical test and age of the patient

We have included in the methods that they were between 6-16 months. As they are individual timepoints there are no statistical tests associated with this data.

Other:

Scale bars are missing in Figs. Supp 1D-G, 2, 4 and Fig.3.

Corrected

Inconsistencies in the text:

-use the same designation throughout the text, such as OVE442 for the mutant mice

Corrected. Changed to *Leat1* mutant throughout the text.

General impression of the manuscript:

The lack of proper statistics combined with weak evidence, overselling of claims and erroneous citations cast doubt on the credibility and seriousness of the study.

We appreciate the comments from the reviewer that has contributed to improving this manuscript. We have also added multiple pieces of evidence to substantiate the conclusions made in the earlier version of this manuscript. This manuscript covers a large amount of work with numerous experiments that contribute to the narrative that a long non-coding RNA *Leat1*, through a potential association with the *EFNB2* gene, drives urethral closure in a hormone-dependent manner. We strongly believe that this body of work as a whole is a significant contribution to the field of urogenital development and our understanding of the development of hypospadias in males.

Reviewer #4 (Remarks to the Author):

The MS describes on a novel hormonally regulated leat1 RNA which binds and facilitates EfnB2 function. Its deletion leads to a suppression of EfnB2 expression with defective urethral closure, hypospadias. In addition, exogenous estrogen suppresses EfnB2 expression.

The work is valued as reporting a novel RNA with its functions for male type urethra formation. It will be also evaluated not only for researchers on male urethra formation but giving impacts on RNA regulation and tissue fusion models.

Some points are puzzling or at least hard to follow by the current form.
I have several points to raise the impacts of MS.

Leat1 and EfnB regulation is the central topic for the MS;

They stated (also in Discussion) as Leat1 binding and facilitating EfnB2 function.
Leat1 binding stabilizes EfnB protein? And it leads to transcriptional autoregulation?
The relationship of leat binding-EfnB and EfnB-autoregulation is puzzling, unclear and should be written clear. They just stated many Ephrins are known to regulate their own expression by one sentence. What is known what is still unclear should be clearly stated.

This has been addressed in the discussion after comments from Reviewer #2.

Leat1 binding to EfnB cytoplasmic region;
Again, what is the background and known examples of RNA binding to EfnB cytoplasmic region and known roles for such regulation? Is this the unique case for leat1? or other examples of RNA binding-mediated auto-regulation?

We agree that these are interesting questions but are beyond the scope of this study. This has not previously been explored and we therefore provide the first evidence of a RNA involved in ephrinb2 signalling that is worthy of further study.

Masculinization, window and induction/suppressive levels ;
Male type urethral formation is the masculinization process and estrogen action is an additional event.
Estrogenic disturbances is intriguing but careful descriptions are necessary for leat1- mediated masculinization.

A critical role for estrogen in masculinisation of the penis has previously been shown (Cripps et al 2024, doi: 10.1038/s42003-024-06048-1; Cripps et al 2019, doi: 10.1016/j.diff.2019.09.001; Govers et al 2019, doi: 10.1096/fj.201802586RR). Misregulation of estrogen during development causes hypospadias, with the data in this manuscript confirming the critical role of estrogen in the process of urethral closure.

Males show rather continuous higher level of expression of leat1 and not showing peaked high level

expression shape in the masculinization window (compared with other type of regulators? Nat Rev Uro ; vol 15, 358–368 (2018)) . Even in such leat1 developmental curve, its “functional “peak exists?

The wording here has been changed to “high levels”.

Related with such masculinization, leat-mediated EfnB induction is significant but looks not robust particularly in vivo (mutants vs WT; rather low degree of induction is observed throughout embryonic stages). This provokes qe that EfnB can be regulated by additional RNAs? And leat1 has other additional targets than EfnB and leat1 ko phenotype can not be solely explained by EfnB? Likewise, the suppressive degree of leat1 and EfnB2 by EE2 is not so prominent. The above points are the feature of RNA mediated masculinization ?(RNA mediated targets regulation ?)

We agree with the reviewer that EfnB2 can be affected by other factors. However, both the Efnb2 reverse signalling mouse and leat1 knockout have hypospadias, therefore, an interaction between the two was the most likely explanation for us to hypothesise. We have expanded the discussion to elucidate this more clearly. Hopefully our further experiments, along with additional comments, have clarified these points.

Conservation among species;

The authors describe on the possible androgen actions of DHT.

They mentioned on the " general conserved model" of Leat1 among species.

and stated the conservation of leat1 extending to marsupials.

Careful descriptions are necessary as they sate the general principle of RNA-EfnB among species;

It is generally known that local production and dependency of DHT varies among species even between mouse-human and Im sure they are experts for such hormonal variation to marsupials.

The DHT used in the GT cultures is required for growth and development of the GT and is not in itself an experimental variable. The conservation of Leat1 among species is a separate experiment and we have been careful to comment only on its genetic conservation, not on its interaction with DHT.

Response to Reviewers

Reviewer #3 (Remarks to the Author):

The authors have addressed several of the concerns raised in the initial review. However, the revised manuscript still suffers from instances of overinterpretation of data, where conclusions extend beyond what the data can reliably support. The following issues remain:

Lines 153–154 and Supplementary Figure 3A:

“...Levels of *EfnB2* in the *Leat1* homozygous male GT were similar to the levels seen during female GT development (Supplementary Fig 3A)...”

“...a) Quantitative real-time RT-PCR showing relative expression of *EfnB2* (mean \pm standard error of mean (SEM)) in the male mutant (black bars, n=4 at each stage) and female wild type (white bars, n=4 at each stage) genital tubercle throughout embryonic development. *EfnB2* expression was, similar to levels seen in the female GT throughout this window...”

This phrasing oversimplifies the actual data. Supplementary Fig. 3A shows significant differences between female and *Leat1* homozygous male GTs at E12.5, E16.5, and E18.5. Therefore, the claim that the expression was “similar throughout this window” is misleading.

The authors should rephrase their statements to more accurately reflect the data.

Our intent here was to show that expression in the *Leat1* homozygous male GT is significantly reduced compared to the WT male GT (Fig 3) and is more similar to the WT female (as shown in Supp 3A). We did not intend for the statement to be misleading. We have rephrased this to specifically address each time point: “Levels of *EfnB2* in the *Leat1* homozygous male GT were similar to the levels seen during female GT development at E13.5, E14.5, E15.5 and E17.5 and significantly reduced compared to female GTs at E12.5.”

The figure legend has also been corrected “*EfnB2* in the *Leat1* homozygous male GT is significantly reduced compared to female WT GTs at E12.5 and is not significantly different at E13.5, E14.5, E15.5 and E17.5.”

Lines 174–182:

“*Leat1* associates with the EPHRINB2 protein” and related Figure 4 with legend

There are still concerns regarding the interpretation of the immunoprecipitation and Western blot data. The description should reflect the experimental evidence precisely. Specifically, if ephrinB2 protein is not directly detected in the Western blot of the

immunoprecipitate obtained with the anti-V5 antibody, it cannot be concluded that ephrinB2 was isolated.

As currently shown in Fig. 4C, the Western blot confirms the presence of the V5 tag in the immunoprecipitated material — not the presence of the ephrinB2 protein itself. To substantiate the claim that ephrinB2 was immunoprecipitated, a Western blot with a specific anti-ephrinB2 antibody is necessary. Alternatively, the text should be rephrased to reflect the actual data.

For example, in the legend:

“..c) Western blot showing the efficiency of protein precipitation with the V5 antibody. EPHRINB2 was isolated specifically using the V5 antibody..”

This is misleading. A more accurate statement would be:

“Western blot showing V5-tag detection in the immunoprecipitate, indicating efficient pull-down of V5-tagged protein.”

Similarly, the description:

“..d) RNA immunoprecipitation showing Leat1 binding to the EPHRINB2 protein...”

is not supported by the data. What is shown is the presence of Leat1 RNA in the immunoprecipitate, which may indicate an interaction but does not demonstrate binding to ephrinB2 unless the presence of ephrinB2 is directly confirmed.

The following phrasing is correct and should serve as a model:

“..Only protein extracts precipitated using the V5 antibody showed presence of the Leat1 transcript...” I strongly encourage the authors to rephrase the relevant text and figure legends to accurately represent the experimental evidence.

As noted in our previous response, we did attempt Western blotting using three different EPHRINB2-specific antibodies. However, none produced reliable results. Therefore, we used a fusion protein of EphrinB2 with a V5 tag—an established and widely accepted molecular approach. In response to the reviewer’s suggestions, we have updated the figure legends and revised the Results section (lines 179–181) accordingly.

Lines 190–322:

Despite the statement of the authors that in vivo mechanisms have been removed from the text, claims suggesting in vivo relevance persist throughout the discussion. These statements should be reviewed carefully and either removed or clearly qualified to avoid misrepresentation of the experimental scope.

We have removed reference to in vivo throughout the discussion in line with the reviewer’s request.

Line 198:

“...Many Ephrins are known to regulate their own expression^{32,33}...”

This statement remains inaccurate. References 32 and 33 pertain specifically to ephrinB1, and do not provide evidence that "many ephrins" self-regulate. Reference 33 cites reference 32 and does not introduce new data on additional ephrin family members. The sentence should be revised to reflect this.

We have revised this to specifically refer to ephrinB1.

Lines 201–205:

“...In the absence of *Leat1*, exogenous *EfnB2* was unable to repress endogenous *EfnB2* expression (Supplementary Fig 3D). This experiment was then repeated in the presence of the cumate inducible *Leat1a* allele described above. There was a significant suppression of mRNA from the endogenous *EfnB2* locus in cells expressing both exogenous *EfnB2* and *Leat1* (Fig 4E). ...”

The description of these findings is difficult to follow because the relevant data are split across two different figures. It would greatly improve clarity if the authors could present these data side by side, as the logical argument hinges on their comparison.

We have removed Supp Fig 3D and included it in Figure 4 as requested.

Additionally, the extent of repression should be described with greater precision. The reduction of endogenous *EfnB2* expression in the absence of *Leat1* but presence of exogenous *EfnB2* is about 20% — this should be stated clearly. In the presence of *Leat1* and exogenous *EfnB2*, the repression is around 50%, which is more substantial, but still not complete. This distinction should be discussed to avoid the impression of exaggerated impact.

The same applies to line 253 in the discussion. I suggest to rather than describing these effects as binary (“repressed” vs. “not repressed”), to characterize the effect more precisely and reflect on the magnitude and biological significance accordingly.

We agree with the reviewer on this point. While we had this data in the figure legend, we now include it in the text as well. “In the absence of *Leat1*, exogenous *EfnB2* only repressed endogenous *EfnB2* expression by approximately 20% (Supplementary Fig 3D). This experiment was then repeated in the presence of the cumate inducible *Leat1a* allele described above. There was a significant suppression of almost 50% mRNA from the endogenous *EfnB2* locus in cells expressing both exogenous *EfnB2* and *Leat1* (Fig 4E).” We have also altered the text at line 253 in the discussion, as suggested: “. In the presence of *Leat1*, exogenous *EfnB2* overexpression was able to suppress transcription of the endogenous *EfnB2* gene by around 50%. However, *EfnB2* transcription was only reduced by approximately 20% in the absence of *Leat1*. Therefore, *Leat1* likely mediates *EfnB2* autoregulation in the developing penis where both genes are co-expressed.”

Reviewer #4 (Remarks to the Author):

I read the revised MS and comments.

About the interaction (Leat1 binding to EfnB2 cytoplasmic region), it is the central issue of this MS.

As also discussed and pointed out by other reviewer, they basically stressed " the current study as the 1st to show Leat1 binding to EfnB" which is true but stating with careful descriptions is very necessary (typically the merits and the limitation of PLA).

We have included the following disclaimer regarding the PLA in the results:

Proximity ligation assays (PLA) are highly sensitive for detecting protein–protein interactions in situ, but are limited in that they cannot determine functional interactions or distinguish between direct and indirect associations in protein proximity. However, when the PLA and V5-tagged-EfnB2 data are taken together, these data suggest that *Leat1* likely associates with the EPHRINB2 protein both in vitro and in vivo

Judged by their reply to my comments (and judged by the technical limits as also discussed with other reviewers), their sum Fig (Fig 7) should be also modified and “softened” including question marks in it. They stated softened sentences in the revision but the current scheme Fig7 is NOT.

We have rewritten both the figure legend and the description of the model to soften the language used. The Figure legend now reads: Proposed model of EfnB2 regulation during mouse urethra formation. For clarity, we have not included question marks on the figure legend but believe that the descriptions have highlighted that the proposed model is our hypothesis of the interactions of *Leat1*.

Also about Masculinization, I agree with the importance of estrogenic actions. But importance of developmental-Masculinization as the Male type urethral formation should be written more as initial backgrounds (currently half or one sentence). Their reply of Cripps paper is intriguing BUT its for EDC and erectile dysfunction.

We have included the following sentences in the introduction:

“Urethra internalization is tightly regulated by hormones and primarily driven by androgens secreted from the fetal testis. In addition, estrogen also plays a critical role

in masculinization of the penis”. The references given here describe the requirement for androgen and estrogen in penis development. Both are critical for correct masculinization of the penis and urethral closure.

Reviewers comments – September 2025

Reviewer #3 (Remarks to the Author):

The manuscript has improved, but it still contains overstatements and overinterpretation. Correlations and co-expression are still presented as though they imply direct interaction or causality. For publication, the text must report the observations objectively and clearly separate evidence from speculation.

In this review, I indicate when applicable the authors' previous responses, and my latest comment.

Line 72

“..Leat1 interacts with EPHRINB2 in the penis..”

Consider rephrasing to more correctly reflect what the data *suggest*.

Changed to “Leat1 associates with EPHRINB2 (either directly or indirectly)..”

Line 74-75

“We further demonstrate that Leat1 is a potential target of endocrine disruption in the etiology of hypospadias in both mice and humans.”

Related to comment on line line 304ff (see further down). No demonstration was provided that Leat1 is a potential target of endocrine disruption in the etiology of hypospadias in humans. Consider rephrasing.

Rephrased: We further demonstrate that *Leat1* expression is impacted downstream of endocrine disruption and hypothesise that *Leat1* could be a potential target of endocrine disruption in the etiology of hypospadias.

Lines 153–154 and Supplementary Figure 3A:

reviewer's comment from 2nd review: “...Levels of EfnB2 in the Leat1 homozygous male GT were similar to the levels seen during female GT development (Supplementary Fig 3A)..”

“...a) Quantitative real-time RT-PCR showing relative expression of EfnB2 (mean \pm standard error of mean (SEM)) in the male mutant (black bars, n=4 at each stage) and female wild type (white bars, n=4 at each stage) genital tubercle throughout embryonic development. EfnB2 expression was, similar to levels seen in the female GT throughout this window...”

This phrasing oversimplifies the actual data. Supplementary Fig. 3A shows significant differences between female and Leat1 homozygous male GTs at E12.5, E16.5, and E18.5. Therefore, the claim that the expression was “similar throughout this window” is

misleading.

The authors should rephrase their statements to more accurately reflect the data.

author's response: "...Our intent here was to show that expression in the *Leat1* homozygous male GT is significantly reduced compared to the WT male GT (Fig 3) and is more similar to the WT female (as shown in Supp 3A). We did not intend for the statement to be misleading. We have rephrased this to specifically address each time point: "Levels of *EfnB2* in the *Leat1* homozygous male GT were similar to the levels seen during female GT development at E13.5, E14.5, E15.5 and E17.5 and significantly reduced compared to female GTs at E12.5."

The figure legend has also been corrected "*EfnB2* in the *Leat1* homozygous male GT is significantly reduced compared to female WT GTs at E12.5 and is not significantly different at E13.5, E14.5, E15.5 and E17.5. ..."

reviewer's latest comment:

Thank you for the clarification and rephrasing. I understand that the intent was to highlight that expression in *Leat1* homozygous male GTs trends more closely to WT female levels than to WT male levels. However, if there are statistically significant differences between mutant males and WT females at several stages, it is problematic to describe the levels as "similar." By definition, significance indicates that the expression levels are not the same, so the claim of similarity becomes misleading. The data in Supplementary Fig. 3A show both reductions (E12.5) and increases (e.g., E16.5, E18.5) compared to WT females. Selectively highlighting only reductions (or time points with no difference) while ignoring significant increases does not accurately reflect the data. A correct description would be that expression in the mutant males shifts toward the female profile but with stage-specific deviations in both directions. As an alternative to comparing day by day, a more global comparison could be considered—e.g., by looking at the overall expression trajectory, or using an aggregate measure (such as area under the curve). This would provide a clearer and scientifically more robust way of showing that the mutant males follow the female profile, even if they do not match it exactly at each stage.

It would improve transparency to comment briefly on the observed high variability—whether it reflects true biological heterogeneity, developmental timing differences, or is potentially influenced by the relatively small sample size (n=4 per group). This would help readers interpret the data more accurately and avoid the impression of selective reporting.

Changed to:

Levels of *EfnB2* in the *Leat1* homozygous male GT trends toward levels of *EfnB2* in the female GT, but with stage-specific deviations in both directions (Supplementary Fig 3A).

We have also included a comment on the variability as requested:

“The variability observed in this data reflects the biological heterogeneity of long non-coding RNAs.”

Lines 190–322:

reviewer's comment from 2nd review: Despite the statement of the authors that in vivo mechanisms have been removed from the text, claims suggesting in vivo relevance persist throughout the discussion. These statements should be reviewed carefully and either removed or clearly qualified to avoid misrepresentation of the experimental scope.

author's response: We have removed reference to in vivo throughout the discussion in line with the reviewer's request.

reviewer's latest comment: There is one more unsupported in vivo claim, in Results: see Line 192. I may have overlooked this before. Why does it take three rounds for fixing this issue?

This reference to in vivo has been removed.

Lines 201–205:

reviewer's comment from 2nd review: ...Additionally, the extent of repression should be described with greater precision. The reduction of endogenous EfnB2 expression in the absence of Leat1 but presence of exogenous EfnB2 is about 20% — this should be stated clearly. In the presence of Leat1 and exogenous EfnB2, the repression is around 50%, which is more substantial, but still not complete. This distinction should be discussed to avoid the impression of exaggerated impact.

The same applies to line 253 in the discussion. I suggest to rather than describing these effects as binary (“repressed” vs. “not repressed”), to characterize the effect more precisely and reflect on the magnitude and biological significance accordingly.

author's response: We agree with the reviewer on this point. While we had this data in the figure legend, we now include it in the text as well. “In the absence of Leat1, exogenous EfnB2 only repressed endogenous EfnB2 expression by approximately 20% (Supplementary Fig 3D). This experiment was then repeated in the presence of the cumate inducible Leat1a allele described above. There was a significant suppression of almost 50% mRNA from the endogenous EfnB2 locus in cells expressing both exogenous EfnB2 and Leat1 (Fig 4E).” We have also altered the text at line 253 in the discussion, as suggested: “. In the presence of Leat1, exogenous EfnB2 overexpression was able to suppress transcription of the endogenous EfnB2 gene by around 50%. However, EfnB2 transcription was only reduced by approximately 20% in the absence of

Leat1. Therefore, Leat1 likely mediates Efnb2 autoregulation in the developing penis where both genes are co-expressed."

reviewer's latest comment:

It seems that the 20% / 50% reductions are treated by the authors as equivalent in impact to full repression, as they only replaced the binary description with percentages. The manuscript would benefit from a brief reflection on the magnitude of these reductions and their potential biological significance.

We have been careful to state the percentage reductions and *not* treat them as equivalent to full repression. At no time have we suggested that there is full repression or that EfnB2 is completely absent. At multiple points in the manuscript we discuss the relevance of reduced *EfnB2* in relation to the degree of severity of hypospadias phenotypes, but are reluctant to correlate levels of repression *in vitro* with biological significance *in vivo*.

Line 183ff

"...To demonstrate a physical interaction *in situ* between Leat1 and the EfnB2 protein we performed Leat1-EfnB2 184 proximity ligation assays (PLA)..."

"...Proximity ligation assays (PLA) are highly sensitive for detecting protein-protein interactions *in situ*, but are limited in that they cannot determine functional interactions or distinguish between direct and indirect associations in protein proximity. However, when the PLA and V5-tagged-EfnB2 data are taken together, these data suggest that Leat1 likely associates with the EPHRINB2 protein both *in vitro* and *in vivo*..."

As I wrote in the first review "...PLA can by its nature only show *in situ* proximity of two partner molecules..." This means close proximity of about 30-40µm. Again, the assay measures proximity, not binding nor physical interaction. Please correct the statements. The data may suggest interaction but do not demonstrate it.

Rephrased to:

"To investigate a physical association *in situ* between *Leat1* and the EfnB2 protein we performed *Leat1*-EfnB2 proximity ligation assays (PLA)..."

We have described the experiment as suggesting an interaction, as requested:

"Although this doesn't demonstrate a direct interaction, when the PLA and V5-tagged-EfnB2 data are taken together, these data suggest that *Leat1* likely associates with the EPHRINB2 protein,..."

Line 301ff:

“...Despite being syntenic with EfnB2, Leat1 overexpression in trans can still drive EfnB2 upregulation, indicating that Leat1 can regulate EfnB2, even outside of its genomic context 54...”

It is not clear which statement/speculation the reference 54 should support. Ref 54 is about long non-coding RNAs in development but does not mention LEAT nor EPHRIN. Consider rephrasing so it becomes clear why the reference is used in this place.

We have clarified that this is in common with other long non-coding RNAs.

Line 304ff:

“...there was variable LEAT1 expression in the UPE of humans with mild hypospadias and an almost complete absence in one third of the cases examined (Fig 6). This, together with the sequence conservation of LEAT1, suggests a conserved interaction with EPHRINB2 to mediate urethra internalization and implicates it as a potential candidate gene in human hypospadias...”

There is an important logical leap: The absence of LEAT1 mRNA in some hypospadias samples does not demonstrate that loss of LEAT1 causes the phenotype. It could be a secondary effect or unrelated. Consider rephrasing.

We have rephrased to:

“This, together with the sequence conservation of *LEAT1*, could suggest a conserved interaction with EPHRINB2 to mediate urethra internalization and implicates it as a potential candidate gene in human hypospadias.”